# *Drosophila* hamlet mediates epithelial tissue assembly of the reproductive system

**Huazhen Wang[1], Ludivine Bertonnier-Brouty[2,3], Isabella Artner[2,3], Jiayu Wen[4]\*, Qi Dai[1]\***

[1]Department of Molecular Bioscience, the Wenner-Gren Institute, Stockholm University, Stockholm, Sweden; [2]Lund University, Lund, Sweden; [3]Lund University Diabetes Center, Lund University, Malmö, Sweden; [4]Division of Genome Sciences and Cancer, The John Curtin School of Medical Research, The Australian National University, and Australian Research Council Centre of Excellence for the Mathematical Analysis of Cellular Systems, Canberra, Australia

## eLife Assessment

This **important** study addresses an essential morphogenetic process-epithelial fusion-by identifying the transcription factor Hamlet as a potential master regulator. Using a combination of genetic, cell biological, and omics approaches, including a comprehensive RNAi screen and high-quality imaging, the authors provide **compelling** evidence for Hamlet's role in coordinating cell fate and differentiation. The findings are robust and of broad interest to developmental biologists and geneticists.

**\*For correspondence:**
jiayu.wen@anu.edu.au (JW);
qi.dai@su.se (QD)

**Competing interest:** The authors declare that no competing interests exist.

**Abstract** Epithelial tissue fusion requires coordinated molecular events at the ends of two epithelial structures. Regulatory mechanisms controlling these events remain largely elusive. In the *Drosophila* reproductive system (RS), this fusion unites the gonad and the genital disc-derived tissues, into a continuous tube. This study unveils the pivotal role of Hamlet (Ham), a *Drosophila* PR domain containing transcription factor, in orchestrating epithelial tissue fusion in the RS. Loss of *ham* leads to sterility and disconnection between the testes and seminal vesicles. Systematic analysis of Ham downstream genes reveals cytoskeletal, metabolic regulators and signaling pathway components. Ham activates genes for epithelial differentiation and remodeling, while repressing genes required for tissue growth and patterning. Using multiplexed in situ hybridization, we demonstrate spatial–temporal gene expression dynamics in contacting epithelia. Key Ham downstream effectors include E-Cadherin (E-Cad), Toll (Tl), and Wnt2 signaling pathways, regulating tissue interaction and fusion. Our findings present a comprehensive gene network crucial for heterotypic epithelial tissue fusion. Mammalian Ham orthologs PRDM3 and PRDM16 are highly expressed in epithelial tissues, suggesting a conserved role across species.

## Introduction

Epithelial tissue fusion is a crucial process for building epithelial tubular network and uniting separate ends (*Bernascone et al., 2017*; *Chung and Andrew, 2008*). Studies from different model systems, like neural tube fusion and *Drosophila* tracheal development, have provided several mechanistic insights into epithelial fusion process. For instance, specialized cells at the leading edge migrate and interact using cytoskeleton protrusions (*Samakovlis et al., 1996a*; *Samakovlis et al., 1996b*; *Sutherland et al., 1996*; *Tanaka-Matakatsu et al., 1996*; *Gervais et al., 2012*; *Jacinto et al., 2001*; *Vasioukhin*

*et al., 2000*) and small GTPases like Rac and Cdc42 (*Pai et al., 2012*). Upon contact, the leading cells re-establish polarity and connections with the help of extracellular matrix molecules, apical proteins and E-Cad (*Caviglia and Luschnig, 2014*). However, the exact regulators and mechanisms controlling epithelial fusion remain to be determined, especially in a context-specific manner.

The reproductive system (RS) in animals comprises tissues from different origins, and must unite to form a connected system for producing and delivering gametes. For example, the human testis is directly linked to a complex tubular system consisting of the efferent and epididymal ducts (*de Mello Santos and Hinton, 2019*). Anomalies in the fusion of these ducts are a significant cause of cryptor-chidism, affecting at least three percent of newborn boys. However, there is little understanding of the molecular mechanisms guiding the patterning and fusion of these ducts.

Similarly, the *Drosophila* male RS (*Figure 1A*) includes the gonad-derived testis and the genital disc (GD)-formed seminal vesicle (SV), accessory gland (AG), ejaculatory duct (EJD), and additional supportive structures. During development, the TE and SV grow toward each other and merge into a continues tubular system (*Jemc, 2011*). This involves two epithelial ends, the terminal epithelial cells in the testis and the SV epithelium. Thus, the developing *Drosophila* RS provides a representative in vivo context to dissect mechanisms underlying epithelial tissue assembly.

Ham belongs to the PR domain containing protein (PRDM) family whose members are evolution-arily conserved cell fate regulators (*Fog et al., 2012*; *Chi and Cohen, 2016*). The PRDM proteins are characterized by the presence of an N-terminal PR (PRDI-BF1 and RIZ1 homology) domain which is similar to the catalytic SET domain shared by histone methyltransferases. Dysfunction of the human orthologs of Ham, PRDM16 (Mel1) and PRDM3 (Evi1, MECOM), has been found in a variety of human abnormalities including congenital diseases, metabolic disorders and cancers (*Bard-Chapeau et al., 2012*; *Zhou et al., 2016*; *Corrigan et al., 2018*; *Cibi et al., 2020*; *Kundu et al., 2020*; *Horvath and Scheele, 2022*; *Wang et al., 2022*; *Hurwitz et al., 2023*). Moreover, in at least two types of epithe-lial cancers (kidney and pancreas cancers), PRDM16 was shown to play a tumor suppressive role (*Kundu et al., 2020*; *Hurwitz et al., 2023*), indicating its importance in maintaining epithelial integ-rity. However, it remains undermined how these PR factors regulate epithelial tissue development and homeostasis.

Ham was exclusively studied in the nervous system where it is required for neuronal dendrite morphogenesis (*Moore et al., 2002*) and cell fate maintenance (*Moore et al., 2004*; *Rives-Quinto et al., 2020*; *Eroglu et al., 2014*). Ham was shown to repress gene expression through modulating chromatin states (*Endo et al., 2012*) or interacting with the co-repressor CtBP (*Moore et al., 2004*; *Endo et al., 2012*). However, the molecular functions of PRDM16 and PRDM3 are versatile, with their PR domains possessing intrinsic histone methyltransferase activity (*Zhou et al., 2016*; *Pinheiro et al., 2012*) and the ability to interact with various co-activators and co-repressors (*Fog et al., 2012*; *Hohenauer and Moore, 2012*). It remains unknown whether Ham plays a role outside of the nervous system and whether Ham also has dual function in gene expression.

In this study, we discovered that Ham is a master regulator in guiding epithelial tissue patterning and fusion in the *Drosophila* RS. Even a slight reduction of Ham disrupts epithelial differentiation and tissue assembly in the RS, ultimately affecting sperm and egg delivery. To explore new mechanisms governing epithelial tissue fusion in the RS, we conducted systematic analyses using RNA-sequencing (RNA-seq), Cleavage Under Targets and Tagmentation (CUT&TAG), multiplexed in situ hybridization data and genetic screening. These analyses led to identification of gene regulatory networks required for epithelial tissue fusion. Ham activates a gene network to promote epithelial cell differentiation and remodeling while suppressing another gene network involved in cell specification and early patterning. These target genes exhibit spatial–temporal expression dynamics during the fusion of the TE and SV. As a key Ham downstream effector, Wnt2 signaling regulates epithelial cell communication and interaction. By elucidating the regulatory network controlled by Ham, our study provides novel insights into the molecular mechanisms underlying epithelial tissue fusion.

## Results
### *Drosophila* RS development is vulnerable to reduced Ham activity
While PRDM16 and MECOM are critical cell fate regulators in various developmental contexts in mammals, the genetic roles of Ham were mainly found in the central and periphery nervous systems

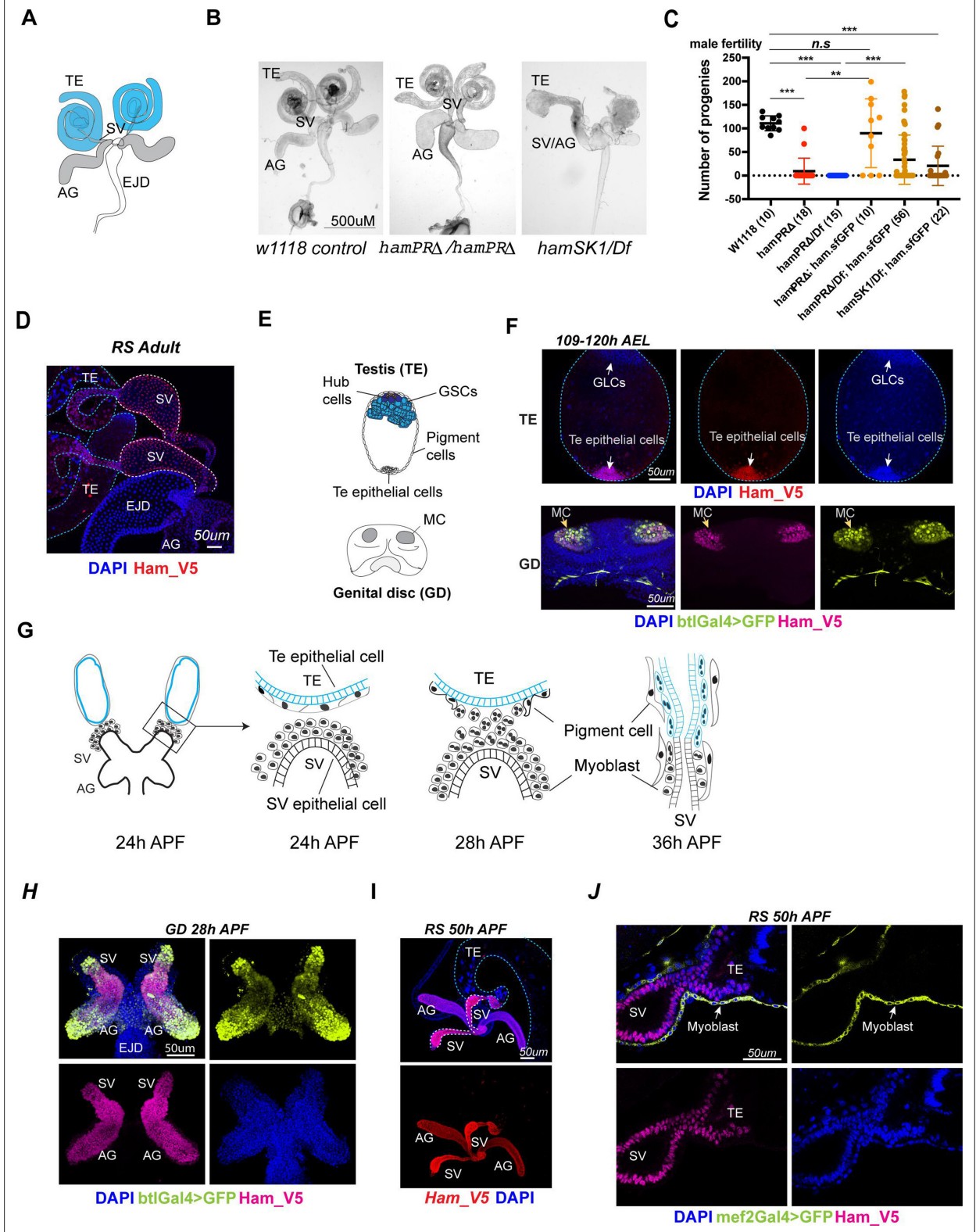

**Figure 1.** *ham* is expressed and required for *Drosophila* male reproductive system (RS) development. (**A**) The adult male RS. TE, testis; SV, seminal vesicle; AG, accessory gland; EJD, ejaculatory duct. (**B**) White-field images of adult male RS, showing defective morphology in the *ham* mutants. (**C**) Quantification of male fertility. Ham_sfGFP with the *ham* genomic region was used to rescue the sterility. Statistical significance was calculated using unpaired *t*-test (***p < 0.001; **p < 0.01; n.s., non-significant). (**D**) Images of the adult RS stained with Ham_V5 in red. DNA is stained with DAPI in blue. The blue and white dashed lines highlight the TE and SV, respectively. (**E**) Illustration of the developing male RS at the larval stage. The testis

*Figure 1 continued on next page*

*Figure 1 continued*

(TE) terminal epithelial cells localize at the opposite side of the germline stem cell (GSC) niche. The genital disc (GD) contains mesenchymal cells (MC) that will develop into the SV and accessory gland (ag). (**F**) Images of the larval TE (top) and GD (bottom) stained with Ham_V5 in red or magenta, the reporter in lime-green (btlGal4 driving UAS-GFP) and DAPI in blue. AEL, after egg laying. (**G**) Illustration of the TE and SV fusion process. APF, after pupal formation. (**H**) Images of the pupal GD stained with Ham_V5 in magenta, the reporter in lime-green (btlGal4 driving UAS-GFP) and DAPI in blue. (**I**) Images of the pupal RS with Ham_V5 in red and DAPI in blue. The blue and white dashed lines highlight the TE and SV, respectively. (**J**) Higher magnification image of the joint site between the SV and TE, with myoblasts and myotubes labeled by mef2Gal4>GFP in lime-green.

The online version of this article includes the following source data and figure supplement(s) for figure 1:

**Source data 1.** Raw counting data for the fertility test in *Figure 1C*.

**Figure supplement 1.** Ham gene locus and mutant alleles.

**Figure supplement 1—source data 1.** This zip archive contains the raw unedited western blot shown in *Figure 1—figure supplement 1C*.

**Figure supplement 1—source data 2.** This zip archive contains the original western blot shown in *Figure 1-figure supplement 1C* with relevant band labeled.

**Figure supplement 2.** Ham expression and function in the female reproductive system (RS).

**Figure supplement 2—source data 1.** Quantification of female fertility shown in *Figure 1—figure supplement 2B*.

(*Moore et al., 2002*; *Moore et al., 2004*; *Rives-Quinto et al., 2020*; *Eroglu et al., 2014*). To systematically study Ham function, we generated new frameshift alleles using CRISPR/Cas9. The sgRNA targets the third exon shared by the three long isoforms B, D and G (*Figure 1—figure supplement 1A*), producing new alleles specifically disrupting the three PR-domain containing isoforms. We therefore named them $ham^{PR\Delta}$. The original $ham^1$ and the null allele $ham^{SK1}$ shifted open reading frame (ORF) for all predicted isoforms (*Moore et al., 2002*; *Rives-Quinto et al., 2020*).

We validated nucleotide deletions in the new alleles by sanger sequencing (*Figure 1—figure supplement 1B*) and protein deletion by western blot using an antibody against Ham. The Ham antibody detected several bands in the blot (*Figure 1—figure supplement 1C*). The two bands at and above the 150 kD marker position in the two heterozygotic controls were not detectable in the $ham^{SK1}$ homozygous mutant, suggesting that these two bands correspond to the longest and the second longest isoforms, respectively. The $ham^{PR\Delta}$ homozygous samples lack the largest band, verifying the depletion of the longest isoforms. We used one of them (–5nt) for all the later experiments.

We compared viability and sensory organ development of these alleles. In contrast to the original report showing that *ham* mutant animals died at the larval stage (*Moore et al., 2002*), we found that $ham^{SK1}$, $ham^{PR\Delta}$, and $ham^1$ alleles all survived to hatch when combined in trans with the *ham* deficiency line (Df) which deletes the entire *ham* locus (*Figure 1—figure supplement 1D*). While $ham^1$/Df and $ham^{SK1}$/Df animals all died right after hatching, about two thirds of the $ham^{PR\Delta}$/Df mutants were viable without obvious gross defects. Furthermore, Ham is known to specify internal cell types (the neuron and sheath) in the mechano-sensory organ lineage, as $ham^1$ mutants exhibited ectopic external cell types (the socket and shaft) with the expense of internal cells (*Moore et al., 2004*). $ham^{PR\Delta}$ mutants displayed normal sensory bristles, in contrast to defective sensory organs in the $ham^1$ and $ham^{SK1}$ alleles (*Figure 1—figure supplement 1E*). Thus, the PR-containing isoforms of Ham are dispensable for mechano-sensory organ development and viability.

However, $ham^{PR\Delta}$ mutant animals showed abnormal RS morphology (*Figure 1B*) and impaired fertility in both male and female animals (*Figure 1C*, *Figure 1—figure supplement 2A, B*). The TE, AG, and SV structures were disrupted in males, while mature eggs were stuck in the mutant female ovaries, indicating failed egg delivery. To confirm that the defects are induced by the *ham* mutation, we performed a rescue experiment by utilizing a pBAC transgene that carries the genomic region of *ham* with the super-folding GFP (sfGFP) coding sequence in frame with the *ham* ORF. The pBAC transgene rescued the viability of all *ham* mutant conditions, and at least partly restored their male fertility (*Figure 1C*). Together, these results indicate that the $ham^{PR\Delta}$ allele represents a hypomorphic condition which preserves Ham shorter isoforms and partial activity. Thus, although viability and sensory organ development do not require the PR-containing isoforms, formation of the RS depends on them.

## Ham-expressing epithelial cells connect the gonadal and supportive tissues

To assess how Ham controls RS development, we examined Ham expression in the developing RS using a V5 antibody which detects all Ham isoforms in the *ham_V5* allele (*Gokcezade et al., 2014*). Ham_V5 was detected in the connecting tube of the adult SV and TE (*Figure 1D*), suggesting a direct involvement of Ham in regulating RS formation.

During RS development, the TE terminal epithelial cells are specified in the embryo as male-specific somatic gonadal precursors (msSGPs) (*DeFalco et al., 2004*). The SV epithelium, together with AG, develops from a group of mesenchymal cells in the GD which undergo mesenchymal-to-epithelial transition (MET) during the larval stage (*Figure 1E*; *Whitworth et al., 2012*; *Ahmad and Baker, 2002*). Notably, in the L3 larval male RS, Ham_V5 was detected in the TE terminal epithelial cells as well as the btl-Gal4-expressing mesenchymal cells in the GD (*Figure 1F*).

In the pupal RS, multiple events, including epithelial tube formation and fusion as well as myoblast and pigment cell migration, are tightly coordinated (*Figure 1G*; *Rothenbusch-Fender et al., 2017*). At this stage, a high level of Ham persists in the TE terminal epithelial cells, SV and AG (*Figure 1H–J*). In contrast, Ham is barely detectable in the SV myoblasts/myotubes labeled by the reporter, mef2-Gal4-driving UAS-GFP (*Figure 1J*). Similarly, high levels of Ham_V5 were found at the joint sites of female oviducts and ovaries during both pupal and adult stages (*Figure 1—figure supplement 2C*). Thus, Ham is an intrinsic factor in epithelial cells at the connection site between the gonadal- and genital-disc-derived tissues in both males and females.

## Ham is required for epithelial tissue fusion in the developing male RS

The specific expression of Ham and mutant phenotypes in the RS prompted us to investigate its function in RS epithelial tissues. Here we focused on the male, as the assembly of the TE and SV represents a remarkable process involving three major cell types (myotubes, epithelial cells, and pigment cells) and a series of tissue reorganization events (*Figure 1G*). As the termini of the TE and SV approach each other, nascent myotubes—formed from myoblasts via cell–cell fusion at the SV surface—make contact with the TE terminus and begin to migrate onto the TE. Simultaneously, epithelial cells at the leading edges converge and fuse to form a continuous epithelial tube. Meanwhile, pigment cells from the TE migrate in the opposite direction. Ultimately, the myotubes organize into a thin muscular sheet, shaping the TE into a characteristic 'spiral' structure (*Bischoff et al., 2021*; *Kuckwa et al., 2016*).

We found that at 40–50 hr APF the TE and SV tubes in *hamPRΔ/Df* and *hamSK1/Df* mutants did not properly elongate and appeared shorter and wider compared to that in the wild-type (*Figure 2A*). The mutant TE appeared as a sphere shape, indicating failed myotube migration. The SV and AG were only partly elongated in the *hamPRΔ* mutant, and even unseparated in the *ham* null allele *hamSK1/Df*. This suggests that the severity of RS abnormality correlates with the levels of reduction in *ham* gene products.

To examine epithelial integrity, we performed immunofluorescent staining using the antibodies against the epithelial cell markers, E-Cad (adherent junctions), and Coracle (Cor, septate junctions equivalent to mammalian tight junction) protein. We also incorporated an SV-specific transgenic reporter, btlGal4-driving UAS-EGFP or UAS-CD8GFP, to the control and *ham* mutant genotypes. E-Cad is enriched at the apical side (the lumen side, *Figure 2B*) of the TE and SV epithelial tubes. Notably, the TE terminal epithelium accumulates a higher level of E-Cad compared to the SV epithelium, while Cor is mainly expressed at the SV side but not much in the TE, suggesting a heterotypic nature of the TE terminal and SV epithelia.

Control animals always showed connected TE and SV, whereas *ham* mutant TE and SV tissues were either separated from each other, or appeared contacted but only with disconnected epithelial tubes (*Figure 2B*, *Figure 2—figure supplement 1A*). Importantly, this phenotype can be fully rescued by the *pBAC_ham_sfGFP* transgene, confirming that it is caused by the *ham* mutation. These observations suggest that the epithelial ends of the TE and SV possess distinct properties despite both being epithelial cell types and that Ham is required for TE and SV fusion.

## Depletion of Ham abolishes TE terminal epithelial cell formation

A close inspection on the pupal TE revealed that most of TE epithelial cells were missing in *ham* mutant animals (*Figure 2B*, *Figure 2—figure supplement 1A*). *ham* mutant lava also either lacked TE

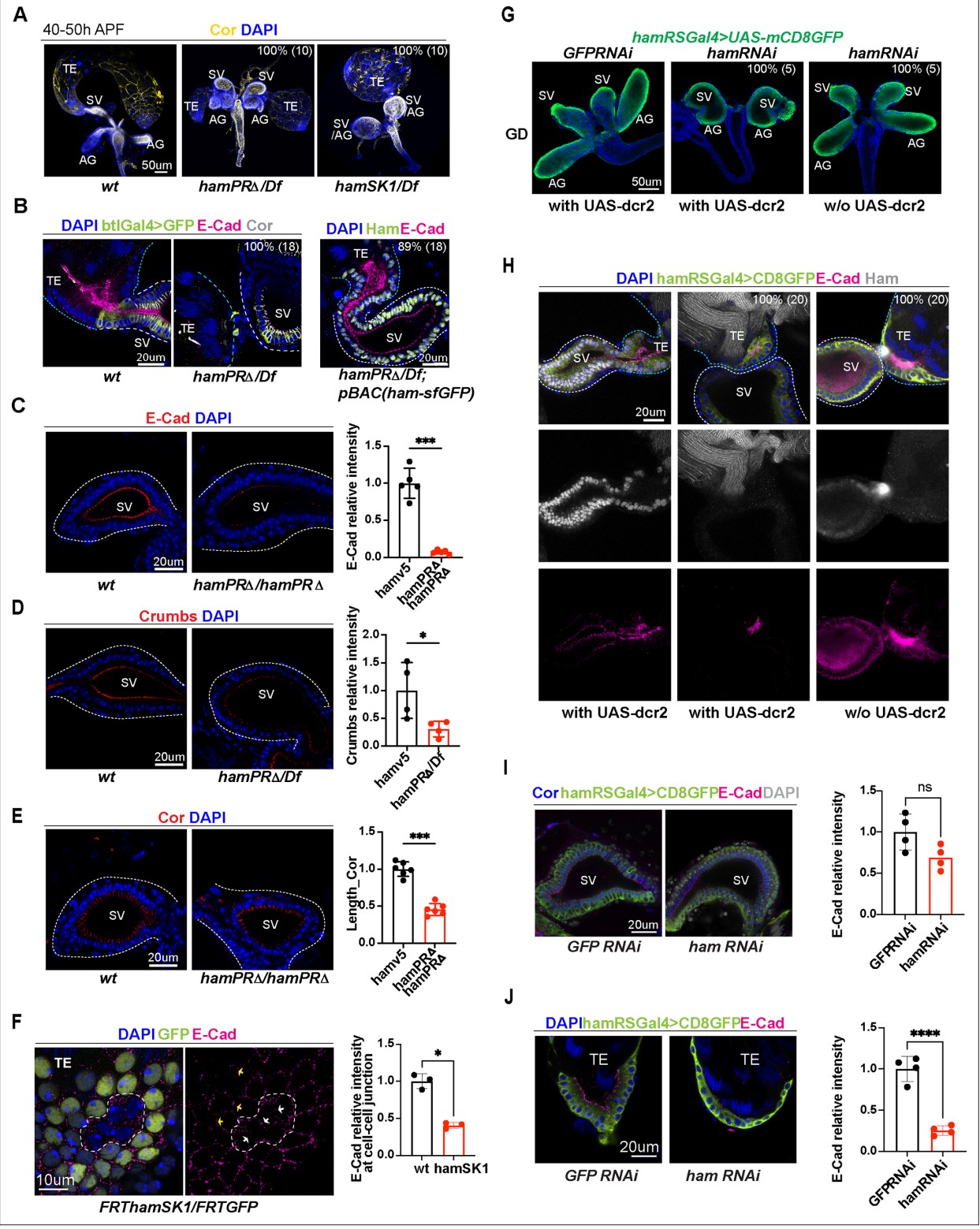

**Figure 2.** Ham promotes differentiation of the TE terminal and seminal vesicle (SV) epithelial cells. (**A**) Images of the male reproductive system (RS) at the pupal stage, showing morphological defects of the TE and SV in the ham mutants. Epithelial cells are labeled with Coracle (Cor). The phenotype penetrance (%) and number of animals (*n*) counted are indicated as % (*n*). (**B**) High magnification images of the connection site of the TE and SV, showing that the ham mutant RS failed to form a continuous tube. Ham-sfGFP restored the normal morphology. Epithelial cells are marked by E-Cad in red, Cor in yellow, and the epithelium on the SV side also marked by btlGal4 driving UAS-mCD8GFP. DNA is labeled by DAPI in blue. (**C**) Images and

*Figure 2 continued on next page*

*Figure 2 continued*

quantification of the E-Cad signal in the *wt* and *hamPRΔ* mutant SV. n = 5. (**D**) Images and quantification of the Crumbs signal in the *wt* and *hamPRΔ / Df* mutant SV. n = 4. (**E**) Images and quantification of the length of Cor signal in the wt and *hamPRΔ* mutant SV. n = 6. (**F**) Images of *ham* mutant mosaic clones in the TE terminal epithelial cells. Note, the mutant cells are GFP negative and within the area surrounded by the white dashed line. The white arrows indicate junctions between mutant cells, while the yellow arrows highlight the junctions between wild-type cells. n = 3. (**G**) Images of pupal male genital disc (GD), showing that *hamRNAi*-mediated knockdown reproduces *ham* mutant phenotypes. (**H**) Images of pupal male RS, stained with E-Cad in red, Ham in yellow, GFP in green and DAPI in blue. UAS-dcr2 enhanced knockdown efficiency and led to more severe morphological defects. (**I**) Images and quantification of E-Cad signal in red and Cor length in blue in control and ham RNAi SV. n = 4. (**J**) Images and quantification of E-Cad signal in control and ham RNAi TE terminal epithelium. n = 4. In the quantification graphs, statistical significance was calculated using unpaired *t*-test except for TE mosaic clones in F which used paired *t*-test (**** p < 0.0001, ***p < 0.001; **p < 0.01; *p < 0.05; ns, non-significant).

The online version of this article includes the following source data and figure supplement(s) for figure 2:

**Source data 1.** Quantification of the E-Cad signal in the *wt* and *hamPRΔ* mutant SV shown in ***Figure 2C***.

**Source data 2.** Quantification of the Crumbs signal in the *wt* and *hamPRΔ /Df* mutant SV shown in ***Figure 2D***.

**Source data 3.** Quantification of the length of Cor signal in the *wt* and *hamPRΔ* mutant SV shown in ***Figure 2E***.

**Source data 4.** Quantification of the E-Cad signal at the border between two *wt* and between two *hamSK1* mutant cells in the TE terminal epithelium shown in ***Figure 2F***.

**Source data 5.** Quantification of E-Cad signal in *GFP* and *ham.RNAi* SV shown in ***Figure 2I***.

**Source data 6.** Quantification of E-Cad signal in *GFP* and *ham.RNAi* TE terminal epithelium shown in ***Figure 2J***.

**Figure supplement 1.** Split channel images for images in ***Figure 2***.

**Figure supplement 2.** Ham controls the formation of TE epithelial cells.

**Figure supplement 3.** An epithelial-specific Gal4 line in the ham locus.

**Figure supplement 3—source data 1.** Quantification of myotube number on the testis in *wt* and *hamPRΔ/Df* shown in ***Figure 2—figure supplement 3C***.

epithelial cells or showed abnormal location of these cells, which can be rescued by the *pBAC_ham_ sfGFP* transgene (***Figure 2—figure supplement 2A***). TE terminal epithelial cells derive from male specific gonadal precursors (mSGPs) specified in the early embryo (***Figure 2—figure supplement 2B***; ***Whitworth et al., 2012***) and expand during larval and early pupal stages. This result indicates that Ham regulates specification or survival of these cells. Indeed, we detected Ham expression in mSGPs and other SGPs in the embryonic gonad. The number of mSGPs labeled by Abd-B or Sox100B-GFP was dramatically reduced in S13 and S15 *ham* mutant embryo (***Figure 2—figure supplement 2C–G***). Our data suggest Ham to be a master regulator for the specification and/or survival of mSGPs, and the absence of mSGPs accounts for loss of TE terminal epithelial cells in larval and pupal stages.

## Ham regulates patterning and cell differentiation of the RS epithelial tubes

The epithelial cells in the SV and AG originate from a group of mesenchymal cells recruited by the FGF pathway involving Btl and Branchless (Bnl) (***Whitworth et al., 2012***; ***Ahmad and Baker, 2002***). These cells undergo MET to form an epithelial mass which then cavitates to generate inner hollow space and eventually elongates into an epithelial tube. In comparison with the control, the signal intensity of both E-Cad and Crumbs (Crb) were significantly reduced in *ham* mutant SV epithelial cells, which coincides with the shortening of septate junctions labeled by Cor (***Figure 2C–E***). This result suggests that Ham regulates epithelial cell differentiation in the SV. To check whether Ham regulates epithelial cell differentiation in the TE, we used the FRT/FLP system (see Methods) to generate mosaic somatic mutant clones. In these animals, *ham* mutant cells were GFP negative, while wild-type and *ham* heterozygous cells expressed GFP (***Figure 2F***, ***Figure 2—figure supplement 1B***). The E-Cad levels at the border between two mutant cells were significantly lower than those between wild-type cells, indicating that *ham* depletion impaired differentiation of TE terminal epithelial cells.

In summary, these phenotypic analyses suggest that Ham controls formation of TE terminal epithelial cells in the early embryo and regulates epithelial tube elongation and cell differentiation at the pupal stage. Thus, failed epithelium fusion in *ham* mutant RS could result from: (1) absence of TE terminal epithelial cells, (2) defects in epithelial tube elongation and cell differentiation of the SV, and (3) a combination of both.

## Ham directly controls epithelial fusion between the TE and SV

Next, we sought to assess whether and how Ham directly regulates the fusion process. We first attempted to generate a mild mutant condition which can bypass defects of TE epithelial cell loss and SV–AG tube elongation. We chose RNAi knockdown because knockdown efficiency can be enhanced by overexpressing Dicer2 (Dcr2) and modulated by temperature when RNAi is controlled by the GAL4/UAS system.

We first identified a Gal4 driver expressed in the SV and AG epithelial cells, after screening the available Gal4 transgenes driven by a series of genomic fragments spanning the *ham* locus (*Jenett et al., 2012*; *Figure 2—figure supplement 3A*). This transgene (thereafter named as *hamRSGAL4*) induces high expression of *UAS-GFP* in the TE and SV epithelial cells and a weak expression in the myoblasts (*Figure 2—figure supplement 3B*). Myotubes appeared at the junction site of the TE and SV and could migrate to the surface of the testis in the *ham* mutant (*Figure 2—figure supplement 3C, D*), confirming that loss of *ham* does not abolishes myoblast fusion or myotube migration. *hamRSGAL4*, when driving a *ham* RNAi line, nearly abolished Ham expression in animals carrying the *UAS-dcr2* transgene (*Figure 2G, H*, the middle panels). These animals also had a reduced number of TE epithelial cells and displayed a ball-like structure of the GD, similar to the *ham* null allele. Without *UAS-dcr2*, *ham* RNAi knockdown reduced Ham expression but preserved TE epithelial cells and normal-appearing SV. Remarkably, the TE and SV epithelia still failed to fuse (*Figure 2H*, the right panel), suggesting that Ham also controls the late step of fusion. Furthermore, E-Cad staining intensity was significantly reduced in *ham* RNAi TE but not in *ham* RNAi SV (*Figure 2I, J*, *Figure 2—figure supplement 1C, D*), suggesting that expression of E-Cad in the TE is sensitive to even mild reduction of Ham protein levels.

## Ham controls distinct molecular pathways to promote tissue assembly

To identify mechanisms that control epithelial fusion, we started to investigate Ham downstream target gene network. We performed RNA-seq experiments from dissected TE and SV tissues during the stage of 18–25 hr APF. We used two pairs of genotypes, including the wild-type (*wt*), and *ham^{PRΔ}* homozygous and the *wt/Df* and *hamSK1/Df* trans-heterozygous (*Figure 3A*). The comparison between the first pair will determine dysregulated genes in the *ham^{PRΔ}* hypomorphic condition, while the comparison between the second pair will identify genes that change expression between the null and *ham* heterozygous conditions.

The principal component analyses validated consistency between the three biological replicates (*Figure 3—figure supplement 1A*). To not exclude small gene expression changes (e.g. in the leading cells which are the minority within the whole tissue), we used a loose cutoff (false discovery rate [FDR] <10% and FC >1). We found more than 2000 up- and downregulated genes in the GD. In the mutant TE, there were similar numbers of dysregulated genes (*Figure 3B*). The genes that only changed expression in one condition could result from unspecific genetic background, differential dependence on the PR-containing isoforms or different sensitivity to *ham* heterozygous condition.

Expression of many dysregulated genes can be indirectly affected by Ham. We next intended to identify genes that are transcriptionally regulated by Ham. To this end, we determined Ham genomic binding sites using Cleavage Under Targets and Tagmentation assays (CUT&TAG) (*Kaya-Okur et al., 2019*) on finely dissected TE and grossly dissected abdominal tissues in which Ham is mainly expressed in the TE and SV epithelium (*Figure 3A*). We used a Ham antibody for wild-type tissues, a GFP antibody for tissues from the *pBAC_ham_SFGFP* transgene and IgG as a negative control in parallel for all the samples. Using the SEACR software, we identified peaks and intersections of peaks across the Ham and GFP antibody conditions (see Methods). We performed principal component analyses and found the GFP antibody gave more consistent peak calling from replicates (*Figure 3—figure supplement 1B*). For subsequent analyses, only the common 3433 and 691 peaks identified by the two antibodies (Ham and GFP) were included to ensure signal specificity (*Figure 3C, D*).

To determine Ham target genes, we intersected the RNA-seq data from the hamSK1/Df versus wt/Df comparison with the Cut&Tag data. Among the 2,546 upregulated genes in the GD, 441 (17.3%) contained Ham CUT&TAG peaks. Among the 2379 downregulated genes, 569 (23.9%) showed Ham binding (*Figure 3E*). In the TE, a lower fraction of dysregulated genes contains Ham CUT&TAG peaks, including 4.7% (108/2274) upregulated genes and 7% (144/2057) downregulated genes (*Figure 3F*), suggesting that dysregulated genes in the mutant TE are mainly indirectly controlled by Ham.

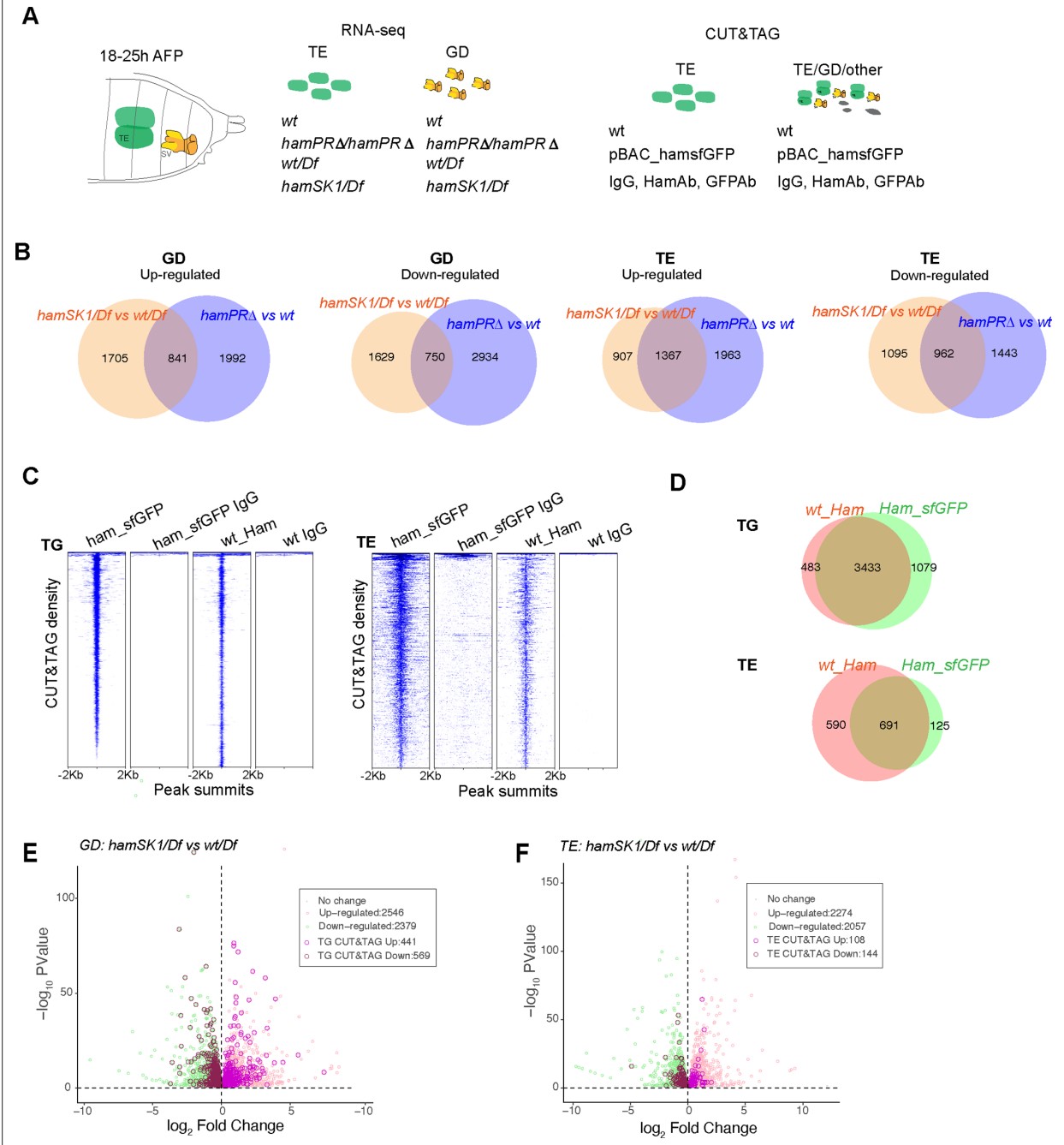

**Figure 3.** Identification of Ham-regulated genes in the developing TE and genital disc (GD). (**A**) Illustration of the sample types and antibodies used in the RNA-seq and CUT&TAG experiments. (**B**) Venn diagram of dysregulated genes between the two mutant conditions in the GD (left) and TE (right) samples. (**C**) Heatmaps of Ham CUT&TAG peak density and IgG control, ranked from high to low and centered at peak summits. (**D**) Venn diagrams of CUT&TAG peaks from the two types of samples. Volcano plots of dysregulated genes in GD (**E**) and TE (**F**) in the *hamSK1/Df* mutant, and the ones with at least one Ham CUT&TAG peak are highlighted.

The online version of this article includes the following source data and figure supplement(s) for figure 3:

**Figure supplement 1.** Ham regulatory activity and potential co-factors.

**Figure supplement 1—source data 1.** Raw data of luciferase activity shown in *Figure 3—figure supplement 1D*.

Both up- and downregulated genes contain Ham binding peaks, suggesting that Ham can repress and activate gene expression. To systematically explore the function of Ham-regulated genes, we performed gene network analysis using STRING (*Szklarczyk et al., 2023*). We focused on the genes bound by Ham and also dysregulated in the genital disc (GD) in the null condition *hamSK1*, considering that most of TE genes are only indirectly influenced in the *ham^{SK1}* mutant. The SRING MSL clustering analysis generated 114 functional clusters from downregulated genes and 62 clusters from upregulated genes (*Figure 4A*, *Supplementary file 1*). Interestingly, many of downregulated genes clustered into several large groups including cytoskeleton regulators, metabolic factors, cell adhesion, and extracellular matrix proteins, suggesting that Ham activates genes to promote epithelial differentiation and remodeling. In contrast, upregulated genes are enriched for patterning factors, cell cycle genes, and transcriptional and translational regulators, suggesting that Ham suppresses early developmental processes. Notably, metabolic factors are enriched in both categories, but these factors belong to distinct functional groups. Thus, epithelial fusion requires complex and balanced metabolic processes. Together, this network analysis confirms that Ham positions on top of distinct molecular pathways to facilitate TE and SV assembly.

Ham was known for its repressive activity via interacting with the co-repressor CtBP and influencing histone modifications (*Endo et al., 2012*). Our finding of its activation function was unexpected. To validate the activation activity of Ham, we chose three downregulated target genes (*Figure 4B*), *dwnt2*, *shot*, and *shg*, for transcriptional reporter assays. *shg* encodes E-cad, while Shot is a member of the spectraplakin family of large cytoskeletal linker molecules simultaneously binding to actin and microtubules. The function of *wnt2* was reported in the developing RS development (*Kozopas et al., 1998*; *Deshpande et al., 2016*). We cloned four genomic fragments from the *wnt2* locus, two each from *shg* and *shot* into a luciferase construct, and co-transfected each reporter with individual plasmids expressing several variants of Ham. In addition to the Ham long isoform and a PR truncation version of Ham, we included the NZF and the CZF domains fused with the VP16 activation domain (NZF_VP16 and CZF_VP16) (*Figure 4C*). The two ZF domains of Ham are putative DNA binding domains, but it remained undetermined whether they help to recruit Ham to DNA. This experimental setting allows to assess which of the ZF domains is able to target Ham-regulated enhancers. The full-length, the PR truncation and the NZF_VP16 fusion all activated expression of the reporters driven by seven out of eight Ham-bound elements (*Figure 4D–G*, *Figure 3—figure supplement 1C, D*). By contrast, the CZF_VP16 had little activity on the reporters. This result confirms that Ham can directly activate target gene expression and that the NZF domain directly or indirectly recruits Ham to enhancers.

To further identify co-factors with which Ham interacts to regulate gene expression, we performed de novo motif discovery from Ham CUT&TAG peaks in mixed TE and GD tissues close to up- and downregulated genes in the *hamSK1* mutant using the MEME-ChIP software. This resulted in nine and five significantly enriched motifs from up- and downregulated gene loci, respectively (*Figure 3—figure supplement 1E*). While five motifs were shared between these two sets, the motifs for Dpp/BMP signaling factor MAD, the zinc-finger protein ZNF263 and two unknown factors were only enriched in the upregulated genes. The result implies that Ham may switch activation to repression activity when interacting with these co-factors in the context of RS development.

## Ham downstream effector genes are required for male RS fertility

Built upon our transcriptomic data, we next intended to identify genes crucial for the fusion process. We reasoned that as the majority of TE terminal epithelial cells are lost in both *ham* mutant alleles, the commonly downregulated genes in *ham^{PRΔ}* and *ham^{SK1}* TE (*Figure 3B*) may be expressed in the *wt* TE epithelial cells and can be functionally important. We thus focused on the top 241 common downregulated genes whose expression showed a fold change greater than 1.5, and the three validated Ham target genes: *shot*, *wnt2*, and *shg*. Using all available RNAi lines from the Bloomington stock center that cover 113 out of 244 candidate genes, we performed a UAS-RNAi screen with the *UAS-dcr2; hamRSGal4* driver for male fertility (see Methods).

The RNAi screen led to identification of 26 genes whose knockdown abolished or reduced male fertility (*Supplementary file 2*). Knocking down four of these genes, *shg*, *AP2-alpha*, *pebble* (*pbl*), and *CG11406*, resulted in complete sterility, indicating their importance for male fertility. We defined this category as 100% penetrance of sterility. Knockdown of other 22 genes reduced fertility to different extent (*Figure 5A, B*). Given that the *hamRSGal4* driver is highly expressed in the TE and SV epithelia,

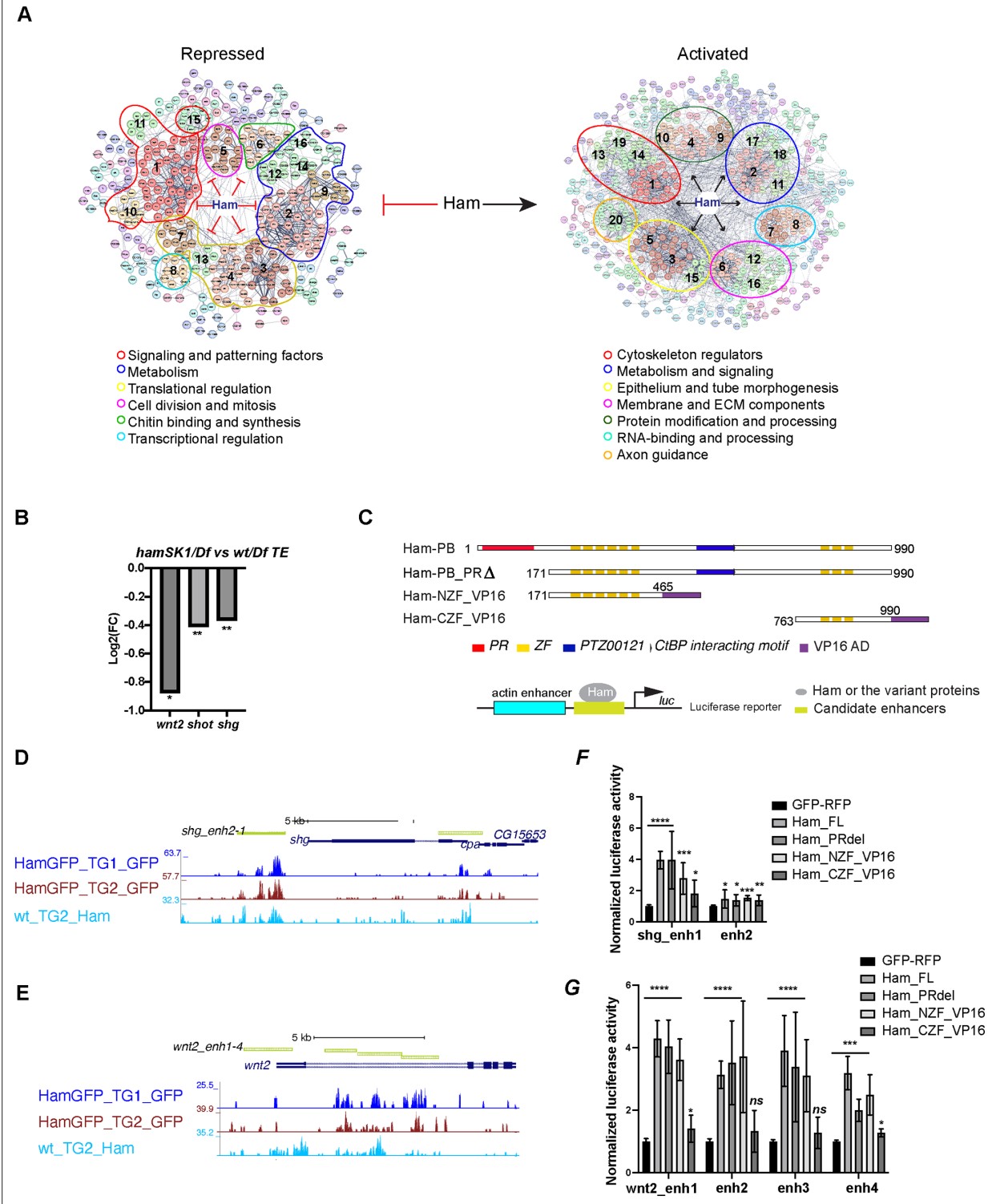

**Figure 4.** Ham activates expression of epithelial differentiation genes. (**A**) Ham gene network analysis from STRING. The position of MCL clusters were manually adjusted to place genes with similar function in close proximity. The functional categories were determined by gene function tools in STRING. (**B**) A bar plot of the log₂ fold-change value from RNA-seq for *wnt2*, *shot*, and *shg*, showing downregulation of these genes in the *hamSK1/Df* mutant. (**C**) Schemes of the Ham_FL and other truncation proteins, and illustration of the luciferase reporter assay. (**D, E**) A genome browser snapshot of the *shg* and *wnt2* loci, showing the CUT&TAG tracks from the TE and GD (TG) mixed samples and the enhancer fragments in the luciferase assays. (**F, G**) Normalized luciferase activity to the empty reporter and then to the control vector that expresses GFP and RFP proteins. Each plot represents

*Figure 4 continued on next page*

*Figure 4 continued*

the average from three biological replicates with each biological replicate having four technical replicates. Statistical significance was calculated using unpaired *t*-test (****p < 0.0001; ***p < 0.001; **p < 0.01; *p < 0.05; ns, non-significant).

The online version of this article includes the following source data for figure 4:

**Source data 1.** Raw data of luciferase activity shown in *Figure 4F*.

**Source data 2.** Raw data of luciferase activity shown in *Figure 4G*.

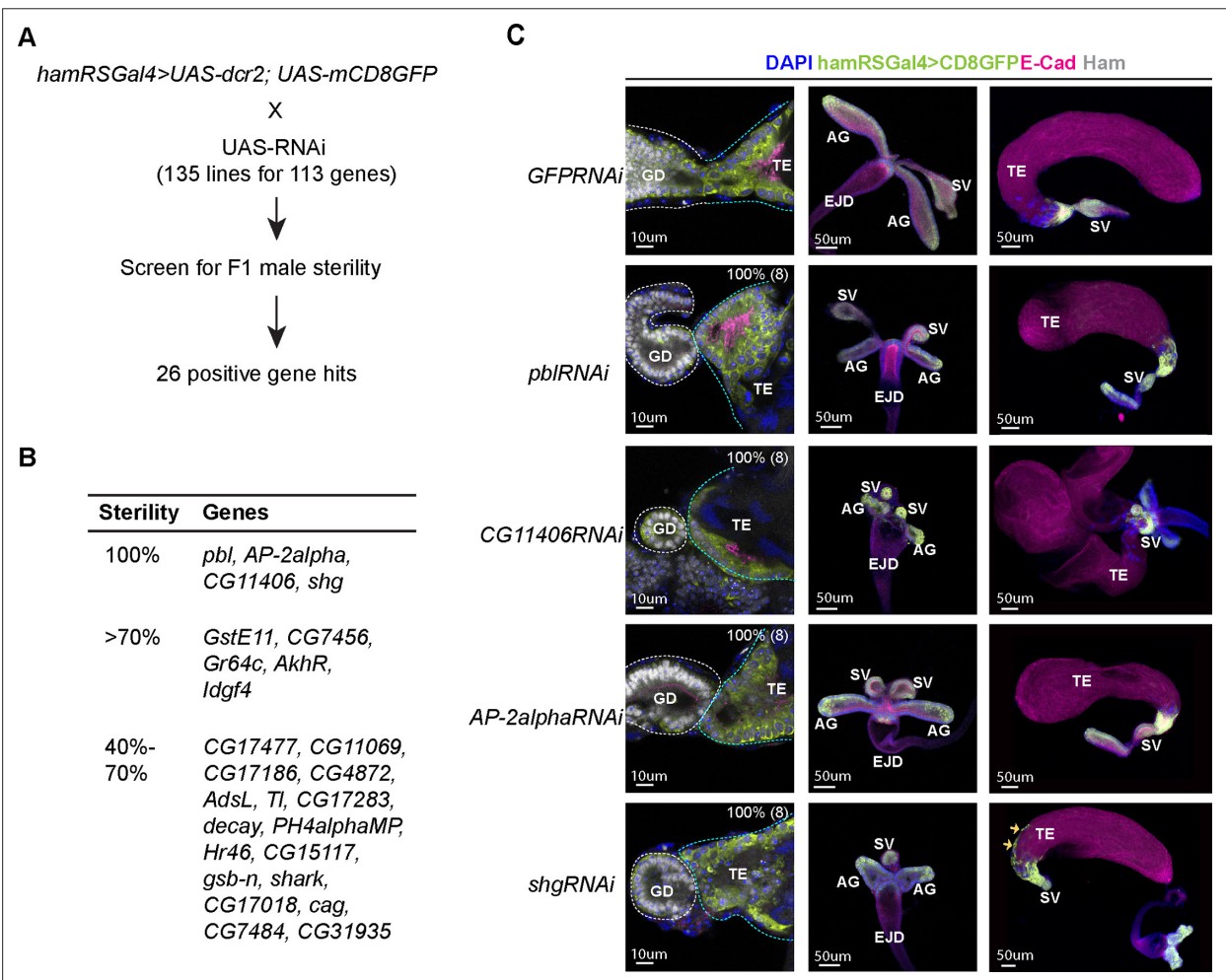

**Figure 5.** A candidate RNAi screen identified Ham-downstream effectors in reproductive system (RS) epithelial tissue fusion. (**A**) Scheme for the RNAi screen. (**B**) Summary of positive hits. The extent of sterility is determined by the number of progenies from the F1 male crossing to the wild-type female (see Methods). 100% means no progeny. >70%, the number of progenies being fewer than 30% of the number of progenies produced by the control cross (the same Gal4 driver × UAS-GFPRNAi). 40–70%, the number of progenies being between 30% and 60% of the number of progenies produced by the control. (**C**) Images of the control and gene-specific RNAi knockdown RS. The left panels show the fusion site stained with E-Cad in magenta, GFP in lime-green, Ham in gray and DAPI in blue, the middle panels are genital disc (GD), and the right ones are TE. The orange arrows indicate mobilized epithelial cells in the *shg* knockdown TE.

The online version of this article includes the following source data and figure supplement(s) for figure 5:

**Source data 1.** Results for a candidate RNAi screen shown in *Figure 5A*.

**Source data 2.** The extent of sterility for positive hits from the RNAi screen shown in *Figure 5B*.

**Figure supplement 1.** Split channel images for images in *Figure 5*.

**Figure supplement 2.** Split channel images for images in *Figure 5*.

we expect highly effective knockdown occurs only in these epithelial cells. However, *hamRSGal4* also drives a weak expression in the myoblasts before they differentiated into myotubes (*Figure 2— figure supplement 3B*), which may result in a non-tissue autonomous effect when knocking down the candidate genes expressed in myoblasts. Furthermore, it cannot be excluded that knockdown in epithelial cells could indirectly influence myotube behavior, which could in turn perturb epithelial morphogenesis.

Consistent with the gene network analysis (*Figure 4A*), these genes include cytoskeleton regulators (*pbl*, *shg*, and *shark*) (*van Impel et al., 2009*; *Oda et al., 1994*; *Ferrante et al., 1995*) and metabolism factors (*CG11406*, *Ph4alphaMP*, *GstE11*, *CG17477*, *CG11069*, *AdSL*, *CG17283*, *CG15117*, and *CG7484*). Other genes are known or predicted to be involved in intracellular membrane trafficking (*AP-2alpha*, *CG31935*, and *CG7456*) (*González-Gaitán and Jäckle, 1997*) and signaling (*wnt2*, *Toll*, *AkhR*, *Idgf4*, and *Hr46*) (*Weber et al., 2003*; *Staubli et al., 2002*; *Kawamura et al., 1999*; *Lam et al., 1997*).

We confirmed that knocking down *pbl*, *AP-2alpha*, *CG11406*, and *shg* led to failed TE and SV fusion and abnormal morphology at the pupal stage (*Figure 5C*, *Figure 5—figure supplements 1 and 2*). *pbl* encodes a Rho guanine nucleotide exchange factor, implicated in multiple processes involving cytoskeleton reorganization (*van Impel et al., 2009*; *Prokopenko et al., 1999*; *Nakamura et al., 2017*). The gene product of CG11406 is a predicted triglyceride lipase, cleaving triacylglycerol to produce diacylglycerol. AP-2alpha, a subunit in the Adaptor complex 2, is involved in intracellular trafficking and secretion (*Lin et al., 2015*). Remarkably, *shg* knockdown also led to aberrant mobilization of TE epithelial cells to the middle of the TE, while no such effect was found in the SV and AG. Downregulation of E-Cad is a key event during epithelial-to-mesenchymal transition, and E-Cad promotes cohesive cell migration of mesenchymal cells, but suppressing cell motility of epithelial cells (*Campbell and Casanova, 2015*; *Chen et al., 1997*). Our result suggests that the SV and the TE terminal epithelia have distinct dependence on E-Cad and that TE epithelial cells are prone to becoming motile when E-Cad is reduced. Together, this screen identified a collection of known and novel factors involved in epithelial tissue fusion, confirming that the fusion process requires coordinated activity of several classes of molecules.

## Ham downstream genes exhibit spatial–temporal expression dynamics in the male RS

During epithelial fusion, cells at the leading edge become specialized and express specific genes required for tissue interaction and remodeling (*Bernascone et al., 2017*; *Chung and Andrew, 2008*). Moreover, as we observed, the TE terminal and SV epithelia have distinct properties, suggesting that these two sets of epithelial cells may express distinct genes. We thus intended to assess spatial distribution of Ham downstream genes. We conducted a high-resolution multiplexed RNA in situ hybridization assay known as SCRINSHOT (single-cell RNA in situ hybridization on tissues) (*Sountoulidis et al., 2020*) to visualize expression of candidate genes. These genes include *wnt2*, *shg*, and *shot* as these genes were either known for male RS development (*wnt2*, *Kozopas et al., 1998*) or epithelial tissue fusion (*shg*, *Lee et al., 2003* and *shot*, *Lee and Kolodziej, 2002*), and the positive genes from the RNAi screen, along with the receptor genes in Wnt2 signaling (*fz*, *fz2*, *fz3*, and *fz4*) and ligand genes in Tl-signaling (*spz*, *spz4*, *spz5*, and *spz6*) (*Supplementary file 3*). Some of these Wnt2 and Tl signaling genes displayed dysregulation in at least one of the *ham* mutant alleles according RNA-seq data (*Figure 6—figure supplement 1A, B*). Other receptor genes in Wnt signaling and ligand genes in Tl signaling were omitted due to low read numbers in the RNA-seq data, indicating negligible or minimal expression in the TE and GD.

Each SCRINSHOT reaction took place within a mini-chamber, accommodating pairs of control and *ham* mutant tissues, or GFP KD control and ham RNAi KD samples (refer to Methods for details of probe design and experimental procedures). This setup was designed to minimize technical variability between the genotypes. mRNAs are visualized as fluorescent dots in SCRINSHOT images, with each dot representing a single mRNA molecule. However, due to the densely packed nature of cells in the TE and SV tissues, quantifying mRNA molecules at a single-cell resolution is challenging. Therefore, we opted to quantify fluorescent signal intensity within defined areas (see Methods). In total, 37 genes were tested, including *ham* itself (*Supplementary file 3*). Probes for seven genes gave either no or low signal in the TE or GD, likely attributed to poor probe quality, given their RNA-seq read numbers

not being low. Resembling Ham protein expression (*Figure 1F*), *ham* mRNAs were detected in the TE terminal epithelium (*Figure 6A*, *Figure 6—figure supplement 2A*), confirming the specificity of the SCRINSHOT method. Similarly, *wnt2*, *shg*, and *shot* all exhibited high expression in the TE terminal epithelium (*Figure 6A*, *Figure 6—figure supplement 2A*).

In total, 25 genes exhibited detectable SCRINSHOT signal in the wt SV and AG, while 29 displayed signals in the wt TE (*Supplementary file 3*). Within the TE, 17 out of 29 were specifically localized in the TE terminal epithelial cells, with the remaining 12 detected only in other TE cells (*Figure 6A*, *Figure 6—figure supplements 2–5*, *Supplementary file 3*).

Subsequently, our analysis focused on the genes expressed in the TE terminal and SV epithelia. While 17 genes exhibited expression in both epithelial ends (*Figure 6A, B*, *Figure 6—figure supplement 2A, B*), 8 were exclusively expressed in SV epithelial cells (*Figure 6F*, *Supplementary file 3*), underscoring the heterotypic nature of the TE and SV epitheliums. Notably, certain genes (CG11406, *AP-2alpha*, *GstE11*, *Tl*, *spz5*, and *spz6*) demonstrated temporal dynamic expression in the TE epithelium: spz5 and CG11406 are present during the 20–25 hr APF timeframe, prior to fusion, but gone during the 28–32 hr APF period when the TE and SV are undergoing fusion; *AP-2alpha*, *GstE11*, *Tl*, and *spz6* showed the opposite trend, being expressed only later (*Figure 6*, *Figure 6—figure supplements 2–4*).

In the GD, while the majority of genes exhibited ubiquitous expression, *wnt2* and *spz* were predominantly expressed in the leading cells within the SV, contrasting with their expression in the remaining GD (*Figure 6B*). Additionally, the expression patterns of *fz2*, *fz3*, and *Tl* displayed spatial specificity: fz2 exhibited higher expression levels in the SV compared to the AG, *fz3* demonstrated the opposite pattern with higher expression in the AG, and *Tl* was primarily expressed in the myoblasts (*Figure 6B*). These findings imply that Wnt2 and Tl signaling may have cell-type-specific roles in the fusion process. Thus, SCRINSHOT mapped 30 candidate genes in the developing GD and TE, with 29 of them display spatial or temporal dynamics.

## Wnt2 and Tl signaling pathways are key mediators of Ham function in RS tissue assembly

Next, we aimed to identify genes exhibiting altered expression in *ham* mutant SV and TE. As *hamSK1* mutant TE lacks terminal epithelial cells, we excluded TE samples and only quantified SCRINSHOT signal levels in the developing SV between *wt* and the *hamSK1* mutant. In contrast, *ham* RNAi knockdown resulted in fusion defects but only mildly impacted SV morphology and TE terminal epithelial cell number (*Figure 2G, H*). These samples allow to assess gene expression alterations in both the SV and TE epithelia. *ham* fluorescent intensity appeared comparable between *wt* and mutant samples (*Figure 6B, D, E*, *Figure 6—figure supplements 2–5*), suggesting that *ham* mutant RNAs, containing the CRISPR-generated indels, remain stable at this stage. In the *hamSK1* mutant SV, seven genes became downregulated, while two exhibited upregulation. The downregulated genes were four Wnt2 pathway genes (*wnt2*, *fz*, *fz2*, and *fz4*), one Tl pathway gene (*spz*), and *shg* (*Figure 6B, C*, *Figure 6—figure supplement 2*). Notably, *wnt2*, *fz*, *spz*, and *shg* also displayed downregulation in *ham* RNAi SV (*Figure 6D, E*, *Figure 6—figure supplement 2*), suggesting that these genes and their associated signaling pathways may be directly involved in the fusion process downstream of Ham. The upregulated genes in *hamSK1* mutant SV include *GstE11* and *CG11069* (*Figure 6—figure supplement 3C, E*), which exhibited low expression in the leading cells of wt SV epithelium, but increased expression under the *hamSK1* condition. In *ham* RNAi knockdown animals, *CG11069* was again upregulated, while *GstE11* did not change expression. Instead, another ligand gene in Tl signaling (*spz5*) displayed increased expression (*Figure 6—figure supplement 3D, F*). On the TE side, only four genes showed expression alteration in *ham* RNAi animals, including *ham* itself, *shg*, *fz*, and *fz2* (*Figure 6—figure supplement 3G*), implicating functional requirement of these genes in both TE and SV epithelia.

The observation that knocking down *Tl* resulted in decreased male fertility (*Figure 5B*, *Supplementary file 2*), coupled with the expression of multiple *spz* ligand genes in the developing TE and SV, suggests that the Tl signaling pathway might be required for assembling the TE and SV. To investigate this possibility, we examined male flies with *spz* knockdown driven by *hamRSGal4*. These animals produced a comparable number of progenies to control animals. The lack of phenotype in this condition could be attributed to inefficient knockdown or genetic compensation by another *spz* gene, like *spz5* whose expression increased in *ham* RNAi animals when *spz* expression was reduced. In addition

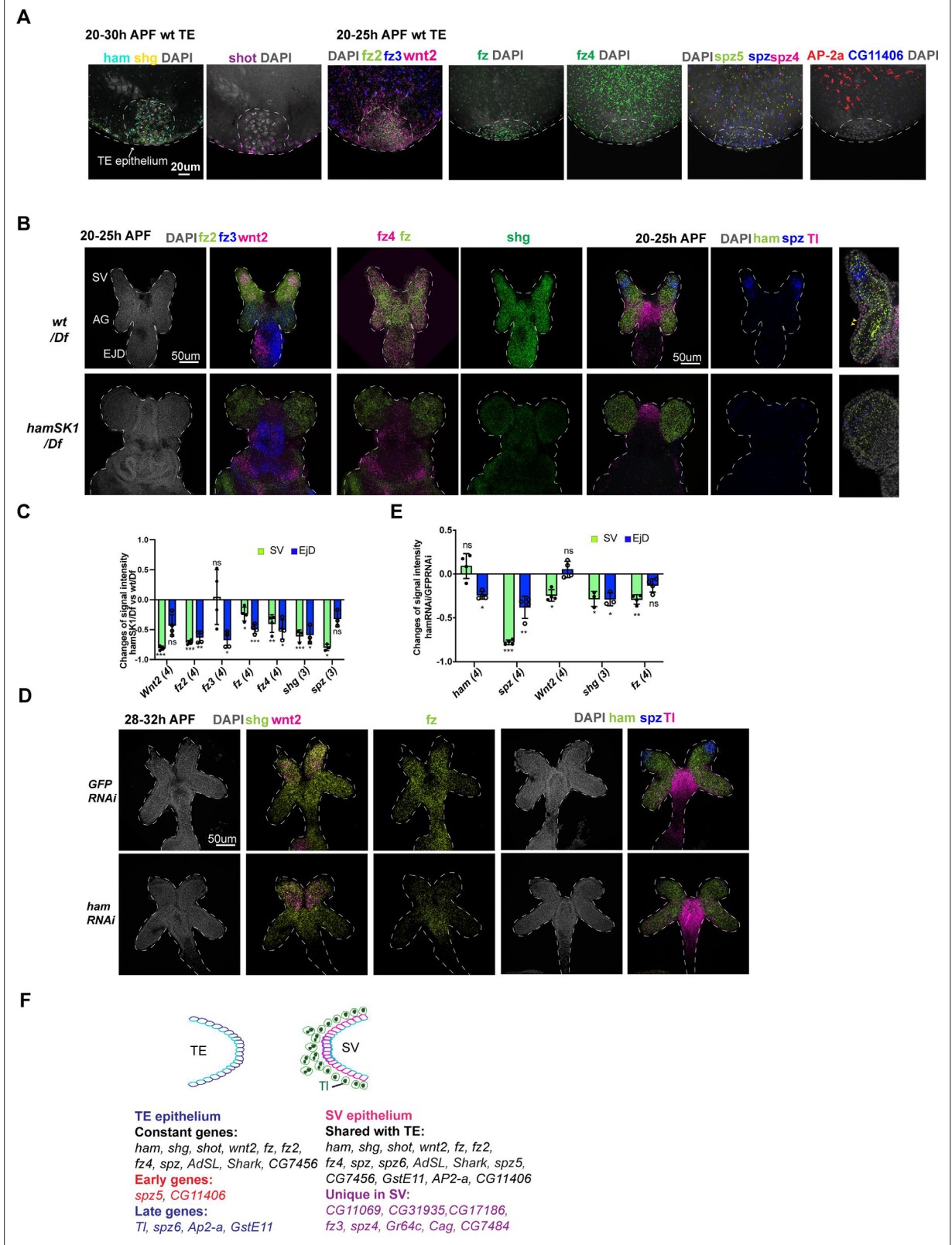

**Figure 6.** Spatial–temporal expression dynamics of Ham downstream genes. (**A**) Images from multiplexed in situ hybridization (the SCRINSHOT method), showing mRNA signal of the indicated genes in *wt* TE samples. DNA is marked by DAPI in gray. (**B**) Images from SCRINSHOT, showing mRNA signals of indicated genes in *wt/Df* and *hamSK1/Df* mutant GD. The last panel is an enlarged view from the sample with *Tl*, *ham*, and *spz* probes. The yellow arrowheads indicate myoblast cells with Tl signal. (**C**) Quantification of signal intensity for genes shown in B in two areas, the SV and EjD.

*Figure 6 continued on next page*

*Figure 6 continued*

Statistical significance was calculated using unpaired *t*-test (***p < 0.001; **p < 0.01; *p < 0.05; ns, non-significant). (**D**) Images from SCRINSHOT, showing mRNA signals of indicated genes in *GFP* and *ham RNAi* GD samples. (**E**) Quantification of signal intensity for genes shown in D in two areas, the SV and EjD. Statistical significance was calculated using unpaired *t*-test (***p < 0.001; **p < 0.01; *p < 0.05; ns, non-significant). (**F**) A summary of genes expressed in the TE terminal and SV epithelium. Highlighted genes show temporal or spatial dynamic expression in the two tissues.

The online version of this article includes the following source data and figure supplement(s) for figure 6:

**Source data 1.** Quantification of SCRINSHOT signal intensity changes for indicated genes between *hamSK1/Df* and *wt/Df* in the areas of SV and EjD, as shown in *Figure 6C*.

**Source data 2.** Quantification of SCRINSHOT signal intensity changes for indicated genes between *hamRNAi* and *GFPRNAi* in the areas of SV and EjD, as shown in *Figure 6E*.

**Figure supplement 1.** Expression of genes tested by SCRINSHOT in RNA-seq data.

**Figure supplement 2.** Split channel images for images in *Figure 6*.

**Figure supplement 3.** Additional genes tested in SCRINSHOT.

**Figure supplement 3—source data 1.** Quantification of SCRINSHOT signal intensity changes for indicated genes between *hamSK1/Df* and *wt/Df* in the areas of SV and EjD, as shown in *Figure 6—figure supplement 3E*.

**Figure supplement 3—source data 2.** Quantification of SCRINSHOT signal intensity changes for indicated genes between *hamRNAi* and *GFPRNAi* in the areas of SV and EjD, as shown in *Figure 6—figure supplement 3F*.

**Figure supplement 3—source data 3.** Quantification of signal intensity for genes that changed expression in *ham RNAi* TE shown in *Figure 6—figure supplement 3G*.

**Figure supplement 4.** Split channel images for images in *Figure 6—figure supplement 3*.

**Figure supplement 5.** Additional split channel images for the images in *Figure 6—figure supplement 3*.

to Tl, two other Tl related genes (Toll-7 and Toll-9) exhibited moderate number of reads in the RNA-seq data. We then tested male fertility in *Toll-7* and *Toll-9* knockdown animals. *Toll-9* knockdown led to a drastic reduction in progeny numbers compared to the RNAi control (*Figure 7—figure supplement 1A*). Consistent with the sterility, some of the *Toll-9* knockdown males displayed abnormal TE shape (*Figure 7—figure supplement 1B*). Remarkably the majority of Toll-9 knockdown animals showed empty SV without sperms (*Figure 7—figure supplement 1C*), suggesting failed connection between the TE and SV. Together, these results pinpoint an essential role of Tl signaling in male RS development.

To validate the role of Wnt2 signaling in RS development, we investigated four genetic conditions, aimed at modulating its level and activity: *wnt2^O* (a *wnt2* null allele), *wnt2* overexpression driven by *hamRSGal4*, *fz*, and *fz2* RNAi knockdown driven by *hamRSGal4*. The *wnt2^O* mutant TE and GD never connect, suggesting that Wnt2 is essential for the fusion process. The *wnt2^O* mutant TE lacks the pigment cell and myocyte layers (*Figure 7A*), consistent with its reported function in these two cell types (*Rothenbusch-Fender et al., 2017*; *Kozopas et al., 1998*). The *wnt2^O* mutant GD failed to form elongated SV and AG, suggesting that Wnt2 is required for GD patterning and differentiation. Moreover, overexpression of *wnt2* also resulted in fusion defects and mis-localization of TE epithelial cells (*Figure 7B*), a phenotype reminiscent of *shg* knockdown animals. Indeed, *wnt2*-overexpressing cells displayed reduced levels of E-Cad, suggesting that Wnt2 signaling might balance E-Cad levels to maintain TE epithelial cell properties. Consistently, *fz2* knockdown animals exhibited comparable phenotypes, including mobilized TE epithelial cells and failed tissue fusion (*Figure 7C*). *fz2* and *fz* knockdown also decreased male fertility (*Figure 7D*). Collectively, these results suggest the importance of optimal Wnt2 signaling levels in male RS development and epithelial tissue fusion.

## Discussion

In this study, we demonstrate that the transcription factor Ham plays a crucial role in controlling epithelial fusion during *Drosophila* RS development. Early in embryonic development, Ham determines the cell fate of msSGPs, which are progenitor cells for TE terminal epithelial cells. Later on, Ham directly regulates the differentiation of epithelial cells in both the TE and SV by orchestrating a gene expression program in epithelial cells. While our primary focus is on male RS, we found that Ham is also expressed in cells at the fusion site of the oviduct and ovary. Mutant females lacking functional

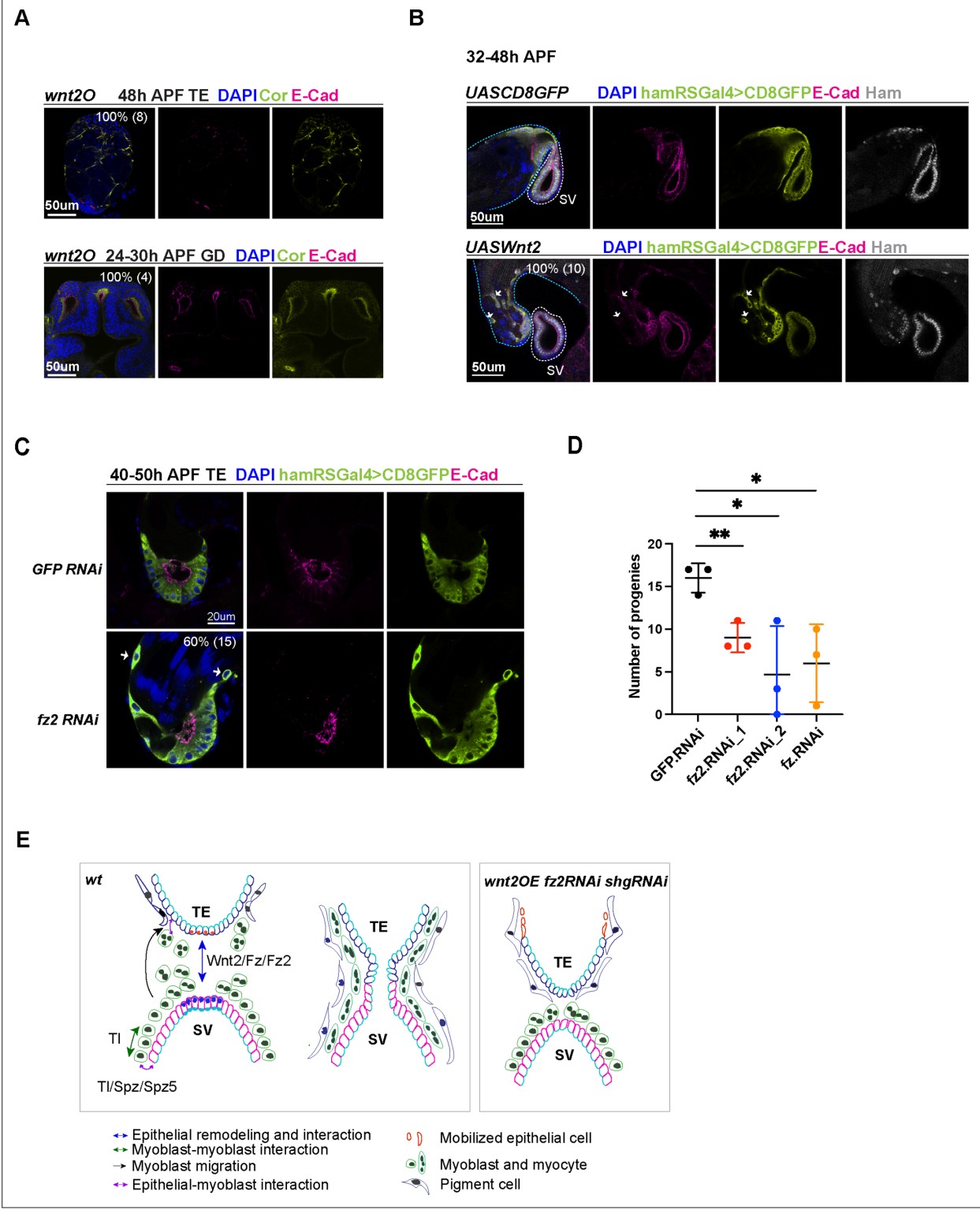

**Figure 7.** Wnt2 and its ligand genes are required for male fertility and reproductive system (RS) development. (**A**) Images of *wnt2o* mutant TE and GD, showing disrupted morphology in these tissues. (**B**) Images of the TE and SV fusion site from the control and *wnt2* overexpression animals. The epithelial cells of the TE and SV are labeled with hamRSGal4 driving UAS-mCD8GFP in lime-green and E-Cad in magenta and Ham in gray. The white arrows indicate mobilized GFP positive cells in the middle of the TE, and these cells have a lower level of E-Cad. The blue and white dashed lines highlight the TE and SV, respectively. (**C**) Images of the TE and SV fusion site from the control and *fz2* RNAi animals. The white arrows indicate mobilized GFP positive cells in the middle of the TE, and these cells have a lower level of E-Cad. (**D**) Graphs of progeny number from male fertility tests for *fz* and

*Figure 7 continued on next page*

*Figure 7 continued*

*fz2* RNAi lines in comparison to GFP RNAi. n = 3. Statistical significance was calculated using unpaired t-test (\*\*, p< 0.01; \*, p<0.05). (**E**) Illustration of cell–cell interactions in the developing RS, and a speculated model for Tl and Wnt2 signaling functions. Wnt2 signaling may be involved in epithelial cell interaction between the two ends. Tl signaling and Tl may be involved in the interaction between epithelial cells and myoblasts/myotubes. In three conditions (*shgRNAi*, *wnt2OE*, and *fz2RNAi*), some TE terminal epithelial cells (illustrated in orange) move up to the distal end of the testis.

The online version of this article includes the following source data and figure supplement(s) for figure 7:

**Source data 1.** Number of progenies from male fertility tests for *fz* and *fz2.RNAi* lines in comparison to *GFP.RNAi* shown in **Figure 7D**.

**Figure supplement 1.** Male fertility test for Toll-9.

**Figure supplement 1—source data 1.** Number of progenies from male fertility test for hamRSGal4 driving *Toll-9 RNAi* in comparison to *GFP RNAi* shown in **Figure 7—figure supplement 1A**.

Ham exhibit a non-functional oviduct epithelial tube, suggesting that Ham likely employs a similar mechanism to control epithelium fusion in females.

By examining *ham* mutant phenotypes and its downstream target genes, we identified several classes of novel factors involved in fusion process of SV and TE. These genes, directly or indirectly controlled by Ham, encompass a diverse array of molecular functions. This discovery provides a comprehensive gene network for future investigation of epithelial fusion processes. The cell-type- and tissue-specific spatial distribution of these genes suggest that they may regulate various aspects of fusion, including cell migration, intercellular communication between myoblasts/myotubes and epithelial cells, specification of the leading cells, and other steps in the fusion process. For instance, Tl and its related receptors is known to mediate cell adhesions (*Umetsu, 2022*). We can speculate that Tl signaling and the molecule itself promote interaction between myoblasts and myoblasts and epithelial cells (*Figure 7E*), based on high expression of Tl in myoblasts and the presence of multiple Spz ligands in the TE and SV epithelial cells. Moreover, *Wnt2*, *fz*, and *fz2* are highly expressed in the leading cells of TE and SV epithelia, and also required for TE and SV connection, suggesting that Wnt2 signaling mediates epithelial cell communication, remodeling, and fusion.

Despite being discovered more than 20 years ago (*Moore et al., 2002*), research on the function of Ham function has primarily focused on its roles in the nervous system. Our findings highlight a crucial function of Ham in epithelial tissues. Apart from the RS, we observed high expression of the Ham protein in other epithelial tissues including the Malpighian tubes (akin to the kidney in insects) and the intestine (data not shown). Furthermore, orthologs of Ham, PRDM16, and PRDM3/MECOM, exhibit high expression in epithelial tubular structures, such as lung, pancreas, kidney, urinary bladder, and gastrointestinal tract and lung, according to observations in the Human Protein Atlas (*Figure 8*, PRDM16, MECOM) (*Karlsson et al., 2021*). Notably, SV and ductus deferens, the equivalent structure to fly SV, are the top one and three tissues with the highest PRDM16 expression (*Figure 8A*). This suggests a potential role for these proteins in controlling epithelial integrity and fusion in mammalian organogenesis. Indeed, we observed that the pancreas duct volume in mouse embryos lacking *Prdm16* is significantly reduced (data not shown). Although the exact cellular changes and the underlying molecular mechanisms in this context await further investigation, this phenotype underscores a crucial function of *Prdm16* in pancreatic epithelial duct formation.

We show that compared to Ham function in the nervous system, the PR-containing isoforms are indispensable for RS epithelial fusion and differentiation, and that Ham can directly activate epithelial gene expression. It remains to be determined whether Ham has distinct molecular functions or it regulates a subset of common molecular pathways in these two contexts. Dendrite morphogenesis involves outgrowth and branching, somewhat analogous to epithelial tube growth and branching. For instance, molecular mechanisms regulating polarized protein trafficking are crucial for both dendrite and epithelial tube branching (*Corty et al., 2009*). Additionally, several transcription factors like Sequoia (*Araújo and Casanova, 2011*; *Brenman et al., 2001*), Tramtrack (*Araújo et al., 2007*; *Li et al., 1997*), and Cut (*Pitsouli and Perrimon, 2013*; *Grueber et al., 2003*) play roles in both dendrite morphogenesis and tracheal tube fusion. These examples suggest that there might be more unrecognized common mechanisms between dendrite and epithelial morphogenesis. It is of interest to test whether the effectors we identified in the RS epithelium are involved in dendrite morphogenesis in the nervous system.

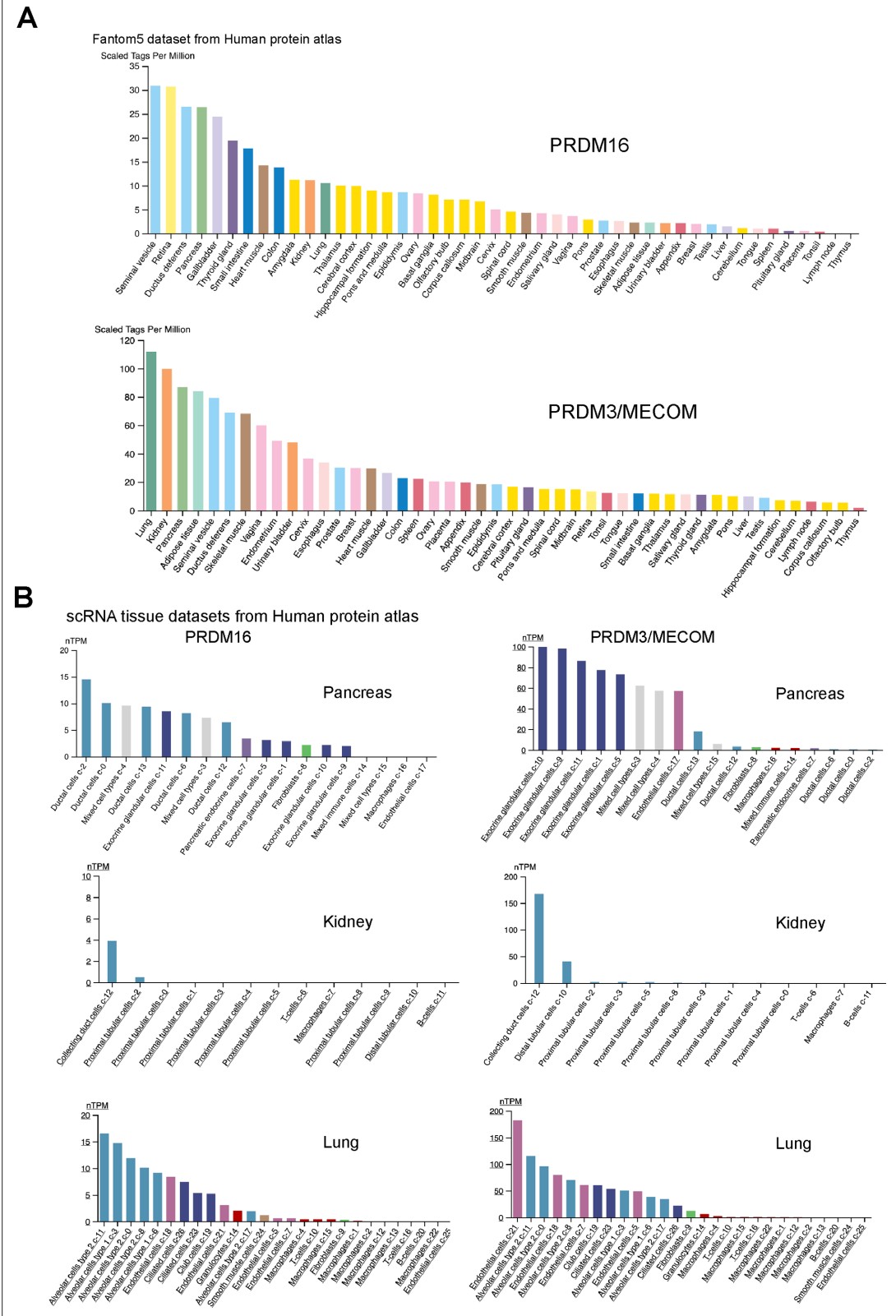

**Figure 8.** PRDM16 and MECOM/PRDM3 expression in human tissues. (**A**) RNA expression data across human tissues through Cap analysis of gene expression (Fantom5 dataset from the Human Protein Atlas). (**B**) Expression in single-cell RNA-seq clusters in the indicated human tissues (from the Human Protein Atlas).

We observed a unique aspect of epithelial fusion in the *Drosophila* RS. Unlike many epithelial fusion processes where two homologous ends merge, such as those in mammalian neural tube closure and *Drosophila* tracheal formation, etc. (*Bernascone et al., 2017*; *Jacinto et al., 2001*; *Pai et al., 2012*), the two epithelial ends in the *Drosophila* RS are heterotypic. Specifically, the TE terminal epithelial cells express higher levels of E-Cad and lower levels of Cor compared to SV epithelial cells. In response to the reduction of E-Cad or ectopic expression of *wnt2*, TE terminal epithelial cells become motile, resembling a process akin to epithelial-to-mesenchymal transition. This suggests that TE terminal epithelial cells may be more invasive during epithelial fusion. On the other hand, SV epithelial cells, despite originating from mesenchymal cells, exhibit typical epithelial cell characteristics, including regular Cor and E-Cad expression. Notably, many genes are expressed exclusively in the SV epithelium, distinguishing it from the TE epithelium. Heterotypic epithelial fusion occurs in many developmental processes, such as digestive system formation, vertebrate optic fissure closure, and mammalian urogenital development. Failure of epithelial fusion can lead to a wide variety of developmental diseases such as oesophageal atresia, spina bifida, ocular coloboma, congenital abnormalities of the kidney, urinary tract, and persistent cloaca. Studying the basic processes and key players of normal epithelial tissue fusion in an in vivo context will help to understand the pathogenesis of these diseases.

## Materials and methods
### Fly strain culturing and generation of transgenes
All fly stocks were maintained at 22°C. The *ham^{PRΔ}* mutants were created using CRISPR: transgenic flies carrying single-guide RNA targeting the PR domain containing isoforms were crossed into the vas-Cas9 transgenic flies. Frame-shift mutations were identified by PCR and Sanger sequencing. Transgenic flies were created at BestGene, Inc.

The ham Gal4 lines and RNAi stocks were obtained from Bloomington Stock Center. Knocking down experiments were performed at 25°C and with a regular 12-hr light–dark cycle.

To generate ham null mutant somatic clones in the TE terminal epithelium, *hamsk1 FRT40A/CTG* males were crossed with virgin females carrying the genotype *hsFLP(X); hsGFP.nls FRT40A*. Embryos aged 0–5 hr were collected and developed at 25°C until the second instar larval stage. GFP-negative larvae, indicating the genotype *hsFLP(X); hsGFP.nls/hamsk1 FRT40A*, were then selected. Head shock treatments at 37°C for 1 hr were administered three times during the second and third instar larval stages to induce somatic clones. Prior to dissection, another 1 hr head shock at 37°C was performed to induce GFP expression, marking the wild-type cells.

### Fertility test
For ham mutants, single male or female with indicated genotype was mated with two w[1118] virgin females or two w[1118] males on day 0 of the experiment. Crosses were flipped every 2 days for three consecutive times. Progenies from the second vial with eggs laid between days 2 and 4 were counted after 2 weeks.

For the RNAi screen, individual UAS-RNAi males were crossed with females carrying UAS-dcr2 and hamRSGal4. The resultant F1 male progenies (2–5 days old) with the genotype UAS-dcr2; hamRS-Gal4>UAS.RNAi were crossed to w[1118] virgin females (1–2 days old) on day 0 of the experiment. In each test, four males were crossed with three females. Crosses were flipped on days 2 and 3 for pre-laying. After flipping on day 3, crosses were kept in the new vial for egg laying for a specific time period (Group A: 48 hr; Group B: 16–18 hr; Group C: 7–8 hr). If at least three males and two females were alive, then the flies were transferred to a new vial, the old vial was kept, and all progenies were counted.

For fertility test of the Wnt2 and Tl signaling pathways components, UAS-RNAi males were crossed to virgin females of the genotype UAS-dcr2; hamRSGal4. Two F1 male progenies (1–2 days old) with the genotype UAS-dcr2; hamRSGal4>UAS.RNAi were crossed to three w[1118] virgin females (1–2 days old) on day 0 of the experiment. Crosses were flipped every day for four consecutive times. Progenies from the fourth vial with eggs laid between days 3 and 4 were counted after 2 weeks.

## Immunostaining

For embryo staining: embryos were collected, dechorionated with bleach, and fixed with 4% form-aldehyde in PEM (0.1 M PIPES, 2 mM EGTA, 1 mM MgSO$_4$, pH 7) for 20 min. For immunostaining, embryos were rehydrated, washed with PBT (0.3% Triton X-100 in PBS), and blocked with PBSBT (0.3% Triton X-100, 1% BSA in PBS).

For tissue staining: tissues were dissected from stage defined larvae or pupae in 1× PBS within 30 min and fixed in 4% paraformaldehyde in PBS for 20 min. For immunostaining, dissected tissues were washed with PBT (0.1% Triton X-100 in PBS) and blocked with PBSBT (0.1% Triton X-100, 1% BSA in PBS).

For both embryo and tissue staining, the primary antibodies were diluted with PBSBT (0.1% Triton X-100, 1% BSA in PBS) and incubated at 4°C over night. After extensive washing with PBSBT (15 min × 4 times), the secondary antibodies were added in 1:500 dilution and incubated at room temperature for 2 hr. After washing with PBT (0.1% Triton X-100 in PBS), the samples were counter-stained with DAPI and mounted in Vectashield (Vector Laboratories).

The following primary antibodies were used: Rabbit anti-Ham (1:300) (Gift from Eric Lai's lab), Ms anti-V5 (1:500, R960-25, invitrogen), chk anti-GFP (1:2000, ab13970, Abcam), Rat anti-vasa (1:50, anti-vasa-c, Developmental Studies Hybridoma Bank, DSHB), Ms anti-ABD-B (1:50, 1A2E9-c), Ms anti-SXL (1:50, M18-c), rat anti-DE-Cad (1:50, DCAD2, DSHB), mouse anti-Crb (1:10, Cq4 DSHB), and mouse anti-Coracle (1:100, C615.16 DSHB). Secondary antibodies conjugated to Cy3 or Cy5 or Alexa Fluor-488 (Jackson ImmunoResearch) were used.

## Quantification of signal intensity from fluorescent immunostaining

For E-Cad and Crb in *Figure 2*, 'raw intensity density' from manually defined regions (using free-hand drawing) along the apical side of the SV and TE terminal epitheliums was obtained in ImageJ. For each sample, two regions in the SV and three regions in the TE terminal epithelium were averaged. The regions used to determine relative E-Cad fluorescence in *hamSK1* TE somatic clones were at the apical side between cell junction. The length of Cor-stained cell junction was measured using ImageJ. A straight line was drawn over the Cor staining with the 'Freehand Line tool'. For each sample, the average length from 10 cell–cell junctions were used. In all experiments, images for the genotypes in comparison were always processed using exact settings. At least three biological replicates were included in statistical analysis.

## Western blot

Embryos from *w1118, hamPRΔ/CTG, hamSK1/CTG* were collected and aged to embryonic stage S13 to S16. In each blot, 50–100st GFP+ or GFP− embryos were used. The selected embryos were homogenized directly in 1xSDS loading buffer (62.5 mM Tris-HCl pH 6.8, 2.5% SDS, 0.002% Bromophenol Blue, 0.7135 M (5%) β-mercaptoethanol, 10% glycerol) and denatured at 98°C heat blot for 5 min. After centrifuged, supernatant was taken for western blotting with 8% of SDS–PAGE. Rabbit anti-Ham (1:2000) primary antibody was used in blotting. To minimize unspecific signal from IgG heavy chains, light chain-specific secondary antibody of mouse anti-Rabbit IgG (1:40,000 Jackson ImmunoResearch) was used, and the signals were developed with ECL Plus reagent (GE Healthcare).

## Cell culture and Luciferase assays

To generate different variants of Ham, the ORFs of HamFL, Ham-NZF-vp16, Ham-CZF-vp16, and *hamPRΔ* were PCR amplified and cloned into the pAC_V5 (nitrogen) vector. The enhancer fragments were cloned in to the p2xTetO-Firefly luciferase vector. All transfections were performed using *Drosophila* S2-R+ cells grown in Schneider *Drosophila* medium containing 10% fetal calf serum. Cells were co-transfected with the Ham expression construct, 2xTetO-Firefly luciferase and pAc-Renilla plasmids in 96-well plate using the Effectene Transfection kit (QIAGEN). Luciferase assays were performed and measured using the Dual Luciferase Assay System (Promega). Expression was calculated as the ratio between the firefly and *Renilla* luciferase activities. Three rounds of transfections were performed on different days and different cell populations and considered as three biological replicates. Within each biological replicate, four wells of cells were transfected with the same DNA mixture and measured and thus considered as four technique replicates. In total, for each DNA combination, the values from the

12 wells of transfection were calculated, averaged and plotted in *Figure 4F, G* and *Figure 3—figure supplement 1D*.

## SCRINSHOT

The *Drosophila* RS was dissected at defined ages in 1× PBS and fixed in 4% paraformaldehyde in PBS for 20 min. After washing with PBST 0.1% (Triton X-100) for 2 × 15 min, fine dissected samples were placed on Poly-lysine coverslip and freezed at –80°C after drying on a heater.

The SCRINSHOT experiments were carried out according to the published method (*Sountoulidis et al., 2020*). In brief, three padlock probes and three corresponding detection probes were designed for each gene of interest. The slides with fine dissected samples were pretreated at 45°C to reduce moisture and fixed in 4% PFA in 1× PBS, followed by washing in PBS Tween-20 0.05% twice. Samples were then dehydrated with 70%, 85%, 100% ETOH and air dry. The SecureSeal hybridization chamber (GRACE BIO-LABS, 621501) was then mounted to cover samples on the slide. Permeabilization of tissues were first done by treating with 10 µg/ml Proteinase K (Invitrogen) in PBSTween-20 0.05% for 7 min followed by washing with glycine (2 mg/ml in PBS Tween-20 0.05%) once and with PBS Tween-20 0.05% for 3 × 5 min. After this, samples were further permeabilized in 0.1 M HCl for 3 min, followed by washing in PBS Tween-20 0.05% twice. Samples were then blocked in a probe-free hybridization reaction mixture of 1× Amplifase buffer (Lucigen, A1905B), 0.05 M KCl, 20% formamide deionized (Millipore S4117), 0.1 µM Oligo-dT, 0.1 µg/µl BSA (New England Biolabs, B9000S), 1 U/µl RiboLock (Thermo, EO0384), and 0.2 µg/µl tRNAs (Ambion, AM7119). Then hybridization of padlock probes was done by incubating samples with padlock probes (with the concentration of each one 0.01 µM) mixed in blocking reagents used before (no oligo dT used in this step). The slide was then put into a PCR machine to denature at 55°C for 15 min and hybridize at 45°C for 120 min. Padlock probes were then ligated by using SplintR ligase (NEB M0375) at 25°C for 16 hr followed by RCA (rolling cycle amplification) at 30°C for 16 hr by using phi29 polymerase (Lucigen, 30221-2) and RCA primer1. Then a fixation step was applied to stabilize RCA product in 4% PFA for 15 min followed by washing in PBS Tween-20 0.05%. Then the hybridization of the first three genes was done by mixing all 3,3′ fluorophore-conjugated detection probes of each gene in reaction reagent 2× SSC, 20% Formamide deionized, 0.1 µg/µl BSA, and 0.5 ng/µl DAPI (Biolegend, 422801) followed by hybridization at 30°C for 1 hr. The slides were washed in 20% formamide in 2× SSC and then in 6× SSC, followed by dehydration in 70%, 85%, and 100% ethanol until the chamber was removed. Then the samples were preserved in SlowFade Gold Antifade mounntant (Thermo, S36936) and kept in dark before imaging. After image acquisition, the first three detection probes were removed by using Uracil-DNA Glycosylase (Thermo, EN0362), and the slides were ready for the next round of detection probe hybridization. The procedure was repeated until all genes were hybridized and imaged. Images were acquired with an airyscan-equipped confocal microscope system (Zeiss LSM 800, Carl Zeiss), and ZEN 2.5 Blue edition software was used for image processing.

## Quantification of SCRINSHOT signals

Similar to quantification of immunofluorescent signal, fluorescence intensity of SCRINSHOT for each gene was measured in defined areas within images of the SV, EjD, and TE in 2D projections. Squares were drawn to define areas at the leading side of the SV, the middle region of the Ejd and the TE terminal. The total fluorescence intensity was first normalized to area size. The resultant value for *ham* mutant or RNAi samples was then contrasted to the normalized intensity of control samples.

## CUT&TAG

CUT&TAG was performed according to the published method (*Kaya-Okur et al., 2019*). In brief, TE and grossly tissue of the posterior abdomen were dissected from 18 to 25 hr APF *w1118* and *pBAC_ham_sfGFP* animals in PBS within 30 min. For grossly tissues, the dissected samples were homogenized in 2 ml NE1 buffer followed by incubation on ice for 10 min. After filtering out the grossly cuticles, nuclei were collected by spinning at 1300 × *g* and resuspended in 1 ml PBS. For TE, nuclei extraction was done by simply pipetted samples in 200 µl NE1 up and down for 20–30 times and incubated on ice for 10 min. Light fixation was performed with 0.1% formaldehyde at RT for 2 min and neutralized with 75 mM glycine. Then the nuclei were washed by PBS and wash buffer one time each and kept in 200 µl wash buffer on ice until enough samples were collected (30–50st animals per/antibody). Resuspended

nuclei in 1.2 ml wash buffer and mixed with 20 µl Concanavalin A-coated magnetic beads (Polyscience 86057) with soft vortexing. Aliquoted samples into two parts immediately and incubated for 10 min with rotation to allow the binding of the nuclei with the beads. The nuclei/beads mix was then blocked with 1 ml cold antibody buffer for 10 min and resuspended in 150 µl cold antibody buffer. For w1118 samples, 1 µl Rabbit anti-Ham or 1 µl Rabbit IgG (Sigma-Aldrich, I5006) was added for an overnight incubation at 4°C. For *pBAC_ham_sfGFP* samples, 1 µl Rabbit anti-GFP (G10362, Invitrogen) or Rabbit IgG was used. The beads/nuclei/antibody mix was then washed with Dig-wash buffer and incubated with secondary antibody (1:100 dilution) for 1 hr. After further washes with the Dig-wash buffer, a pA-Tn5 adaptor complex in Dig-300 buffer was added to the beads for 1 hr reaction. The beads were then washed with Dig-300 buffer before the incubation with 300 µl Tagmentation buffer at 37°C for 1 hr. Then 10 µl 0.5 M EDTA, 3 µl 10% SDS, and 2.5 µl 20 mg/ml proteinase K (Invitrogen) were used to stop the reaction. Then the resultant DNA was purified with DNA clean & Concentrator kit (ZYMO Research D4013) and eluted in 25 µl Elution buffer. To generate libraries, 21 µl fragment DNA was mixed with 2 µl 10 µM Universal i5 primer, 2 µl 10 µM uniquely barcoded i7 primer and 25 µl PCR master mix (NEB Phusion High-Fidelity PCR Master Mix with HF Buffer, M0531). The PCR condition was as follows: 72°C 5 min, 98°C 30 s, repeat 12 times (98°C 10 s, 63°C 10 s), 72°C 1 min and hold 4°C. The libraries were cleaned with standard Ampure XP beads as previously described. Libraries from four biological replicates were produced for each condition.

We mapped the CUT&TAG samples of *w1118* and *pBAC_ham_sfGFP* in TE and TG tissues to the dm6 genome assembly using Bowtie2 with the following parameters: bowtie2 −−end-to-end −−very-sensitive −−no-mixed −−no-discordant −−phred33 -I 10 -X 700. The coverage for these samples was generated using bedtools genomeCoverageBed and normalized to library depth to yield reads per million per base. For peak calling, we employed SEACR (*Meers et al., 2019*), which is tailored for sparse chromatin profiling data like CUT&TAG. We first used a relaxed FDR cutoff of <0.5 for each replicate (SEACR_1.3.sh normalized_coverage 0.5 norm stringent output_peaks), we then identified common peaks across replicates for each condition. To select confident peaks for *w1118* and *pBAC_ham_sfGFP*, we overlapped their peaks within TE and TG separately, choosing the top 1800 peaks from TE and the top 6000 from TG. Peak annotation was performed by HOMER's annotatePeaks.pl. Peak overlap analysis was performed using HOMER's mergePeaks function with the default -d given parameter, ensuring that only directly overlapping peaks were merged. This method preserved the integrity of the original peak boundaries and maintained the specified distances between peaks from the initial peak calls. De novo motif discovery for peaks in up- or downregulated genes, positioned 2000 bp upstream and 500 bp downstream of the TSS in TE and TG tissues, was performed using the MEME-ChIP software. The parameters used were: -ccut 100, -meme-p 5, -dna, -meme-mod anr, -minw 5, -maxw 15, and -filter-thresh 0.05.

## RNA-seq

Total RNA was extracted from finely dissected GDs or testis samples from 18 to 25 hr APF animals using Trizol reagent (Invitrogen). RNA quality was tested by Agilent Bioanalyzer.

RNA-seq libraries were made using the Illumina Truseq Total RNA library Prep Kit LT. Sequencing was performed on the Illumina Hiseq2500 platform. Following the trimming of adapter sequences with Trimmomatic, RNA-seq reads from replicated wild type (×3) and mutant samples (×3) were aligned to the *Drosophila melanogaster* (dm6) genome assembly utilizing HISAT2. Gene annotations were sourced from UCSC. The DESeq2 Bioconductor package was employed to identify differentially expressed mRNAs between *w1118* and the *ham^{PRΔ}*/*ham^{PRΔ}* mutant, as well as between +/Df and *ham^{SK1}*/Df mutant, with a subsequent estimation of the FDR. Genes with an FDR <10% and absolute $\log_2$FC >0 were considered differentially expressed. Details of the calculated differential expression are provided in *Supplementary file 4, table 4*.

## STRING gene network analyses

Ham activated (genes downregulated in *hamSK1* and bound by Ham in TG) and repressed (genes upregulated in *hamSK1* and bound by Ham in TG) gene networks are constructed using String, MCL clustering find natural clusters the stochastic flow, inflation parameter, 2. Disconnected genes are hidden. The position of the clusters is manually adjusted to fit related GO categories. The links to the gene networks in STRING: https://string-db.org/cgi/network?taskId=bbF8IgMTqf1M&

sessionId=bzDIAZvMgWsi; https://string-db.org/cgi/network?taskId=bhKCcMoLlbwh&sessionId=bzDIAZvMgWsi

## Acknowledgements

We thank the imaging facility (IFSU) of Stockholm University, the BEA sequencing facility at Karolinska Insitute, and the National Genomic Institute of Scilife Laboratories, Sweden, for providing service and support. The Bloomington Stock Center and the *Drosophila* community provided important reagents and fly stocks. We thank Christos Samakovlis's lab for helping to set up SCRINSHOT and Eric Lai's lab for the initial Ham reagents. Funding from Australian Research Council Centre of Excellence for the Mathematical Analysis of Cellular Systems (CE230100001) and ANU Future Scheme to JW The project was supported by project grant from Swedish Research Council (Vetenskapsrådet, 2020-03543) and Swedish Cancerfonden (CAN 22 2161 Pj) to QD.

## Additional information

### Funding

| Funder | Grant reference number | Author |
|---|---|---|
| Vetenskapsrådet | VR 2020-03543 | Qi Dai |
| Cancerfonden | CAN 22 2161 Pj | Qi Dai |
| ARC Centre of Excellence for the Mathematical Analysis of Cellular Systems (MACSYS) | CE230100001 | Jiayu Wen |
| Australian National University Future Scheme | | Jiayu Wen |

The funders had no role in study design, data collection, and interpretation, or the decision to submit the work for publication.

### Author contributions

Huazhen Wang, Data curation, Formal analysis, Validation, Investigation, Visualization, Methodology, Writing – review and editing; Ludivine Bertonnier-Brouty, Isabella Artner, Investigation; Jiayu Wen, Data curation, Software, Investigation, Writing – review and editing; Qi Dai, Conceptualization, Resources, Supervision, Funding acquisition, Investigation, Writing – original draft, Project administration, Writing – review and editing

### Author ORCIDs

Huazhen Wang http://orcid.org/0000-0002-4875-1829
Ludivine Bertonnier-Brouty https://orcid.org/0000-0003-1294-3244
Isabella Artner https://orcid.org/0000-0002-2895-5664
Jiayu Wen http://orcid.org/0000-0003-1249-6456
Qi Dai https://orcid.org/0000-0002-2082-0693

### Ethics

Drosophila work (Dnr 5.5 18-07072/2024) from the Swedish Board of Agriculture.

Reviewer #1 (Public review): https://doi.org/10.7554/eLife.104164.3.sa1
Reviewer #2 (Public review): https://doi.org/10.7554/eLife.104164.3.sa2
Author response https://doi.org/10.7554/eLife.104164.3.sa3

## Additional files

### Supplementary files

Supplementary file 1. Clusters and genes from STRING network analysis.

Supplementary file 2. Genes that showed male sterility in the RNAi screen.

Supplementary file 3. Tested genes and their expression in SCRINSHOT.

Supplementary file 4. Differentially expressed genes from RNA-seq data.

MDAR checklist

## Data availability

All sequencing data produced in this study have been submitted to Gene Expression Omnibus (accession number: GSE290310). All newly produced materials from this study are available for the research community.

The following dataset was generated:

| Author(s) | Year | Dataset title | Dataset URL | Database and Identifier |
|---|---|---|---|---|
| Dai Q, Wen J, Wang H | 2025 | *Drosophila* Hamlet mediates epithelial tissue assembly of the reproductive system | https://www.ncbi.nlm.nih.gov/geo/query/acc.cgi?acc=GSE290310 | NCBI Gene Expression Omnibus, GSE290310 |

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

# Appendix 1

## Appendix 1—key resources table

| Reagent type (species) or resource | Designation | Source or reference | Identifiers | Additional information |
|---|---|---|---|---|
| Genetic reagent (*D. melanogaster*) | w1118 | Bloomington *Drosophila* Stock Center | BDSC:3605 FLYBase: FBst0003605; RRID:BDSC_3605 | FlyBase symbol: *w1118* |
| Genetic reagent (*D. melanogaster*) | hamPR Δ FRT40A/Cyo | This paper | | A new hamlet mutant fly line |
| Genetic reagent (*D. melanogaster*) | hamSK1 FRT40A/Cyo | This paper | | A new hamlet mutant fly line |
| Genetic reagent (*D. melanogaster*) | ham1,FRT40A/Cyo | **Moore et al., 2002** DOI: 10.1126/ science.1072387 | | |
| Genetic reagent (*D. melanogaster*) | pBAC_ham_sfGFP | Bloomington *Drosophila* Stock Center | BDSC:83660 FLYBase: FBst0083660; RRID:BDSC_83660 | FlyBase symbol: *y1 w*; PBac {ham-GFP.FPTB}VK00033/TM6C, Sb1* |
| Genetic reagent (*D. melanogaster*) | hamRSGal4 | Bloomington *Drosophila* Stock Center | BDSC:48361 FLYBase: FBst0048361; RRID:BDSC_48361 | FlyBase symbol: *w1118;P {GMR80G06-GAL4}attP2* |
| Genetic reagent (*D. melanogaster*) | hamRNAi | Bloomington *Drosophila* Stock Center | BDSC:26728 FlyBase: FBst0026728; RRID:BDSC_26728 | FlyBase symbol: *y1 v1;P{TRiP.JF02270}attP2* |
| Genetic reagent (*D. melanogaster*) | UAS-GFP | Bloomington *Drosophila* Stock Center | BDSC:4775 FlyBase: FBst0004775; RRID:BDSC_4775 | FlyBase symbol: *w1118; P{UAS-GFP.nls}14* |
| Genetic reagent (*D. melanogaster*) | GFP.RNAi | Bloomington *Drosophila* Stock Center | BDSC:9330 FlyBase: FBst0009330; RRID:BDSC_9330 | FlyBase symbol: *w1118;P{UAS-GFP.RNAi.R}142* |
| Genetic reagent (*D. melanogaster*) | Wnt2O | Bloomington *Drosophila* Stock Center | BDSC:6958 FlyBase: FBst0006958; RRID:BDSC_6958 | FlyBase symbol: *Wnt2O/CyO, amosRoi-1* |
| Genetic reagent (*D. melanogaster*) | UASWnt2 | Bloomington *Drosophila* Stock Center | BDSC:6961 FlyBase: FBst0006961; RRID:BDSC_6961 | FlyBase symbol: |
| Genetic reagent (*D. melanogaster*) | fz2.RNAi | Bloomington *Drosophila* Stock Center | BDSC:27568 FlyBase: FBst0027568; RRID:BDSC_27568 | FlyBase symbol: *y1 v1;P{TRiP.JF02722}attP2* |
| Genetic reagent (*D. melanogaster*) | fz2.RNAi | Bloomington *Drosophila* Stock Center | BDSC: 67863 FlyBase: FBst0067863; RRID:BDSC_67863 | FlyBase symbol: *y1 sc* v1 sev21; P{TRiP.HMS05675} attP40* |
| Genetic reagent (*D. melanogaster*) | fz.RNAi | Vienna *Drosophila* Resource Center | VDRC: 43075 FlyBase: FBst0464905; | FlyBase symbol: *w1118;P{GD4614}v43075* |
| Genetic reagent (*D. melanogaster*) | Toll-9.RNAi | Bloomington *Drosophila* Stock Center | BDSC: 34853 FlyBase: FBst0034853; RRID:BDSC_34853 | FlyBase symbol: *y1 sc* v1 sev21; P{TRiP.HMS00171} attP2* |
| Genetic reagent (*D. melanogaster*) | pbl.RNAi | Bloomington *Drosophila* Stock Center | BDSC: 28343 FlyBase: FBst0028343; RRID:BDSC_28343 | FlyBase symbol: *y1 v1;P{TRiP.JF02979}attP2* |
| Genetic reagent (*D. melanogaster*) | CG11406.RNAi | Bloomington *Drosophila* Stock Center | BDSC: 61185 FlyBase: FBst0061185; RRID:BDSC_61185 | FlyBase symbol: *y1 v1;P{TRiP.HMJ22905}attP40* |
| Genetic reagent (*D. melanogaster*) | AP-2alpha.RNAi | Bloomington *Drosophila* Stock Center | BDSC: 32866 FlyBase: FBst0032866; RRID:BDSC_32866 | FlyBase symbol: *y1 sc* v1 sev21; P{TRiP.HMS00653} attP2* |
| Genetic reagent (*D. melanogaster*) | shg.RNAi | Bloomington *Drosophila* Stock Center | BDSC: 32904 FlyBase: FBst0032904; RRID:BDSC_32904 | FlyBase symbol: *y1 sc* v1 sev21; P{TRiP.HMS00693} attP2* |
| Genetic reagent (*D. melanogaster*) | GstE11.RNAi | Bloomington *Drosophila* Stock Center | BDSC: 82975 FlyBase: FBst0082975; RRID:BDSC_82975 | FlyBase symbol: *y1 sc* v1 sev21; P{TRiP.HMC06653} attP40* |

*Appendix 1 Continued on next page*

*Appendix 1 Continued*

| Reagent type (species) or resource | Designation | Source or reference | Identifiers | Additional information |
|---|---|---|---|---|
| Genetic reagent (*D. melanogaster*) | CG7456.RNAi | Bloomington *Drosophila* Stock Center | BDSC: 51866 FlyBase: FBst0051866; RRID:BDSC_51866 | FlyBase symbol: *y1 v1;P{TRiP.HMC03440}attP40* |
| Genetic reagent (*D. melanogaster*) | CG7456.RNAi | Bloomington *Drosophila* Stock Center | BDSC: 53679 FlyBase: FBst0053679; RRID:BDSC_53679 | FlyBase symbol: *y1 v1;P{TRiP.HMJ21592}attP40* |
| Genetic reagent (*D. melanogaster*) | Gr64c.RNAi | Bloomington *Drosophila* Stock Center | BDSC: 36734 FlyBase: FBst0036734; RRID:BDSC_36734 | FlyBase symbol: *y1 sc* v1 sev21;P{TRiP.HMS01625}attP40* |
| Genetic reagent (*D. melanogaster*) | AkhR.RNAi | Bloomington *Drosophila* Stock Center | BDSC: 29577 FlyBase: FBst0029577; RRID:BDSC_29577 | FlyBase symbol: *y1v1;P{TRiP.JF03256}attP2* |
| Genetic reagent (*D. melanogaster*) | AkhR.RNAi | Bloomington *Drosophila* Stock Center | BDSC: 51710 FlyBase: FBst0051710; RRID:BDSC_51710 | FlyBase symbol: *y1v1; P{TRiP.HMC03228}attP40* |
| Genetic reagent (*D. melanogaster*) | Idgf4.RNAi | Bloomington *Drosophila* Stock Center | BDSC: 55381 FlyBase: FBst0055381; RRID:BDSC_55381 | FlyBase symbol: *y1 sc* v1 sev21; P{TRiP.HMC04069}attP40* |
| Genetic reagent (*D. melanogaster*) | CG17477.RNAi | Bloomington *Drosophila* Stock Center | BDSC: 42821 FlyBase: FBst0042821; RRID:BDSC_42821 | FlyBase symbol: *y1 v1;P{TRiP.HMS02503}attP40* |
| Genetic reagent (*D. melanogaster*) | CG11069.RNAi | Bloomington *Drosophila* Stock Center | BDSC: 77158 FlyBase: FBst0077158; RRID:BDSC_77158 | FlyBase symbol: *y1 sc* v1 sev21; P{TRiP.HMS05930}attP40* |
| Genetic reagent (*D. melanogaster*) | CG17186.RNAi | Bloomington *Drosophila* Stock Center | BDSC: 27079 FlyBase: FBst0027079; RRID:BDSC_27079 | FlyBase symbol: *y1 v1;P{TRiP.JF02425}attP2* |
| Genetic reagent (*D. melanogaster*) | CG4872.RNAi | Bloomington *Drosophila* Stock Center | BDSC: 64486 FlyBase: FBst0064486; RRID:BDSC_64486 | FlyBase symbol: *y1 v1;P{TRiP.HMC05503}attP40* |
| Genetic reagent (*D. melanogaster*) | AdSL.RNAi | Bloomington *Drosophila* Stock Center | BDSC: 34347 FlyBase: FBst0034347; RRID:BDSC_34347 | FlyBase symbol: *y1 sc* v1 sev21; P{TRiP.HMS01336}attP2* |
| Genetic reagent (*D. melanogaster*) | Tl.RNAi | Bloomington *Drosophila* Stock Center | BDSC: 31044 FlyBase: FBst0031044; RRID:BDSC_31044 | FlyBase symbol: *y1 v1;P{TRiP.JF01491}attP2* |
| Genetic reagent (*D. melanogaster*) | CG17283.RNAi | Bloomington *Drosophila* Stock Center | BDSC: 63723 FlyBase: FBst0063723; RRID:BDSC_63723 | FlyBase symbol: *y1 v1;P{TRiP.HMJ30291}attP40/CyO* |
| Genetic reagent (*D. melanogaster*) | decay.RNAi | Bloomington *Drosophila* Stock Center | BDSC:65879 FlyBase: FBst0065879; RRID:BDSC_65879 | FlyBase symbol: *y1 sc* v1 sev21; P{TRiP.HMC06141}attP2* |
| Genetic reagent (*D. melanogaster*) | PH4alphaMP.RNAi | Bloomington *Drosophila* Stock Center | BDSC: 65190 FlyBase: FBst0065190; RRID:BDSC_65190 | FlyBase symbol: *y1 sc* v1 sev21; P{TRiP.HMC06065}attP40* |
| Genetic reagent (*D. melanogaster*) | Hr46.RNAi | Bloomington *Drosophila* Stock Center | BDSC: 27253 FlyBase: FBst0027253; RRID:BDSC_27253 | FlyBase symbol: *y1 v1;P{TRiP.JF02542}attP2* |
| Genetic reagent (*D. melanogaster*) | CG15117.RNAi | Bloomington *Drosophila* Stock Center | BDSC: 33693 FlyBase: FBst0033693; RRID:BDSC_33693 | FlyBase symbol: *y1 sc* v1 sev21; P{TRiP.HMS00562}attP2* |
| Genetic reagent (*D. melanogaster*) | gsb-n.RNAi | Bloomington *Drosophila* Stock Center | BDSC: 28078 FlyBase: FBst0028078; RRID:BDSC_28078 | FlyBase symbol: *y1 v1;P{TRiP.JF02915}attP2* |
| Genetic reagent (*D. melanogaster*) | shark.RNAi | Bloomington *Drosophila* Stock Center | BDSC: 25788 FlyBase: FBst0025788; RRID:BDSC_25788 | FlyBase symbol: *y1 v1;P{TRiP.JF01794}attP2* |
| Genetic reagent (*D. melanogaster*) | CG17018.RNAi | Bloomington *Drosophila* Stock Center | BDSC: 57024 FlyBase: FBst0057024; RRID:BDSC_57024 | FlyBase symbol: *y1 sc* v1 sev21; P{TRiP.HMS04468}attP40* |

*Appendix 1 Continued on next page*

*Appendix 1 Continued*

| Reagent type (species) or resource | Designation | Source or reference | Identifiers | Additional information |
|---|---|---|---|---|
| Genetic reagent (*D. melanogaster*) | *cag.RNAi* | Bloomington *Drosophila* Stock Center | BDSC: 63554 FlyBase: FBst0063554; RRID:BDSC_63554 | FlyBase symbol: *y1 v1;P{TRiP.HMJ30120}attP40/CyO* |
| Genetic reagent (*D. melanogaster*) | *CG7484.RNAi* | Bloomington *Drosophila* Stock Center | BDSC: 61869 FlyBase: FBst0061869; RRID:BDSC_61869 | FlyBase symbol: *y1 v1;P{TRiP.HMJ23358}attP40* |
| Genetic reagent (*D. melanogaster*) | *CG31935.RNAi* | Bloomington *Drosophila* Stock Center | BDSC: 64495 FlyBase: FBst0064495; RRID:BDSC_64495 | FlyBase symbol: *y1 sc* v1 sev21; P{TRiP.HMC05513} attP40* |
| Antibody | Rabbit anti-Ham | Gift from Eric Lai's lab | | IF (1:300), WB (1:2000) Cut&Tag (1:150) |
| Antibody | Rabbit anti-GFP | Invitrogen | Cat# G10362 | Cut&Tag (1:150) |
| Antibody | IgG from rabbit serum | Sigma-Aldrich | Cat# I5006-10MG | Cut&Tag (1:150) |
| Antibody | guineapig anti-rabbit | Antibodies online | Cat# ABIN101961 | Secondary antibody for Cut&Tag (1:100) |
| Antibody | Mouse anti-V5 | Invitrogen | Cat# R960-25 | IF (1:500) |
| Antibody | chicken anti-GFP | Abcam | Cat# ab13970 | IF (1:2000) |
| Antibody | Rat anti-vasa | Developmental Studies Hybridoma Bank, DSHB | Cat# anti-vasa-c | IF (1:50) |
| Antibody | Mouse anti-ABD-B | DSHB | Cat# 1A2E9-c | IF (1:50) |
| Antibody | Mouse anti-SXL | DSHB | Cat# M18-c | IF (1:50) |
| Antibody | Rat anti-E-Cad | DSHB | Cat# DCAD2 | IF (1:50) |
| Antibody | Mouse anti-Crb | DSHB | Cat# Cq4 | IF (1:10) |
| Antibody | Mouse anti-Cor | DSHB | Cat# C615.16 | IF (1:100) |
| antibody | Secondary antibodies conjugated to Cy3 or Cy5 or Alexa Fluor-488 | Jackson ImmunoResearch | | IF (1:1000) |
| Antibody | mouse anti-Rabbit IgG light chain specific | Jackson ImmunoResearch | Cat# 211-032-171 | WB (1:40,000) |
| Sequence-based reagent | hamPRΔ_CRISPR gRNA | This paper | gRNA sequence for CRISPR/Cas9 | TCAGGACGATTGTATAGGCG |
| Sequence-based reagent | hamPRΔ check_Forward | This paper | PCR primers | AACACTTTTG AGGGTTGTGTCTTG |
| Sequence-based reagent | hamPRΔ check_Reverse | This paper | PCR primers | ACAGGACTCTTGGGTCGCCC |
| Sequence-based reagent | DWnt2_Luc1_F | This paper | PCR primers | TTGGCGCGCCACTCCTCCC TCCATTTGGCCAC |
| Sequence-based reagent | DWnt2_Luc1_R | This paper | PCR primers | TTGGCGCGCCACCTTCTTC TGCTCCAGCCAACG |
| Sequence-based reagent | DWnt2_Luc2_F | This paper | PCR primers | TTGGCGCGCCGCTTGAAAAG GATCCGTGATC |
| Sequence-based reagent | DWnt2_Luc2_R | This paper | PCR primers | TTGGCGCGCCCCATCATTGATGGAAGGTC |
| Sequence-based reagent | DWnt2_Luc3_F | This paper | PCR primers | TTGGCGCGCCCTCGCAGTGTCG TACATATGTC |
| Sequence-based reagent | DWnt2_Luc3_R | This paper | PCR primers | TTGGCGCGCCTGAGATCACATG CAATGACGC |
| Sequence-based reagent | DWnt2_Luc4_F | This paper | PCR primers | TTGGCGCGCCGCGTCATTGCAT GTGATCTC |
| Sequence-based reagent | DWnt2_Luc4_R | This paper | PCR primers | TTGGCGCGCCGCGGCAATAATCA TGCGATC |
| Sequence-based reagent | shot_Luc1_F | This paper | PCR primers | TTGGCGCGCCAGTGCATGGATG CTAGGCGAAC |

*Appendix 1 Continued on next page*

*Appendix 1 Continued*

| Reagent type (species) or resource | Designation | Source or reference | Identifiers | Additional information |
|---|---|---|---|---|
| Sequence-based reagent | shot_Luc1_R | This paper | PCR primers | TTGGCGCGCCAACCAGCTTGT GCCTGCTACG |
| Sequence-based reagent | shot_Luc2_F | This paper | PCR primers | TTGGCGCGCCAGGTGTATAGA GCGGTGCATG |
| Sequence-based reagent | shot_Luc2_R | This paper | PCR primers | TTGGCGCGCCCAGTGCGCTTT ATATCGCTCG |
| Sequence-based reagent | shg_Luc1_F | This paper | PCR primers | TTGGCGCGCCCAACTCGAACT CGACTCAGTGG |
| Sequence-based reagent | shg_Luc1_R | This paper | PCR primers | TTGGCGCGCCGTGTCGCTGG AACTCTTCCTTG |
| Sequence-based reagent | shg_Luc2_F | This paper | PCR primers | TTGGCGCGCCCTGCCACGAAT GTTGACGATCC |
| Sequence-based reagent | shg_Luc2_R | This paper | PCR primers | TTGGCGCGCCCGATCTGACGA CGACAGATGTC |
| Sequence-based reagent | ham_PL1 | This paper | SCRINSHOT padlock probe | /5Phos/ATCCTGTATATCAAGATGATTCCTCT ATGATTACTGACTGCGTCTATTTAGTGGAGC CGCCCCTATCTTCTTTGGTGAAGTAGCTCCTG |
| Sequence-based reagent | ham_PL2 | This paper | SCRINSHOT padlock probe | /5Phos/ATGGTAGTTTCTGATCAAAGTCCTCT ATGATTACTGACTGCGTCTATTTAGTGGAGCC GCCCCTATCTTCTTTTCACCCTTTAGAAAGCCAA |
| Sequence-based reagent | ham_PL3 | This paper | SCRINSHOT padlock probe | /5Phos/CGAAGTACTCCTCCCTTTCCTCTATGATT ACTGACTGCGTCTATTTAGTGGAGCCGCCCCTATCT TCTTTCATGAACTTCCTTATGTCAT |
| Sequence-based reagent | wnt2_PL1 | This paper | SCRINSHOT padlock probe | /5Phos/AGCTCATTAGTGACGCAATTTCCTCTATGAT TACTGACTGCGTCTATTTAGTGGAGCCGCCCC TATCT TCTTTGCGGAGATTAACATAAATGC |
| Sequence-based reagent | wnt2_PL2 | This paper | SCRINSHOT padlock probe | /5Phos/TATCGCCAACCAGTCGAATCCTCTATGAT TACTGACTGCGTCTATTTAGTGGAGCCGCCCCTAT CTTCTTTTTGATACTTCAGCATGAGTC |
| Sequence-based reagent | wnt2_PL3 | This paper | SCRINSHOT padlock probe | /5Phos/GACATAGCCCTAGTTCTAGTCCTCTATGAT TACTGACTGCGTCTATTTAGTGGAGCCGCCCCTAT CTTCTTTCTTAATCAACGCATTTCCTAT |
| Sequence-based reagent | shg_PL1 | This paper | SCRINSHOT padlock probe | /5Phos/ATCGAATCGAACTCGTACATCCTCTATGAT TACTGACTGCGTCTATTTAGTGGAGCCGCCCCTAT CTTCTTTCAACCGCTCAATAATTATCA |
| Sequence-based reagent | shg_PL2 | This paper | SCRINSHOT padlock probe | /5Phos/CGATTATCGCATACGATTGGTCCTCTATGA TTACTGACTGCGTCTATTTAGTGGAGCCGCCCCTA TCTTCTTTACTACGTTATATGCCCGCA |
| Sequence-based reagent | shg_PL3 | This paper | SCRINSHOT padlock probe | /5Phos/TTACGTTGATGAATCGGATGTCCTCTATGA TTACTGACTGCGTCTATTTAGTGGAGCCGCCCCTA TCTTCTTTCTGATAAACTCCTCCTTGG |
| Sequence-based reagent | shot_PL1 | This paper | SCRINSHOT padlock probe | /5Phos/TCGCGGTAATCCTTGCATCCTCTATGATT ACTGACTGCGTCTATTTAGTGGAGCCGCCCCTAT CTTCTTTTATCAATAGTTGGTGCATAG |
| Sequence-based reagent | shot_PL2 | This paper | SCRINSHOT padlock probe | /5Phos/TCCTTTAGGGTCTTCATTACTCCTCTATG ATTACTGACTGCGTCTATTTAGTGGAGCCGCCC CTATCTTCTTTCAAGATAGTGTTTCCTCTAGA |
| Sequence-based reagent | shot_PL3 | This paper | SCRINSHOT padlock probe | /5Phos/TATGCCATCGATGAAGACATTCCTCTAT GATTACTGACTGCGTCTATTTAGTGGAGCCGC CCCTATCTTCTTTATGTATCGAATTTCGTATTGAG |
| Sequence-based reagent | pbl_PL1 | This paper | SCRINSHOT padlock probe | /5Phos/TCGTAAACGTATAGAGCTTCTCATGATT ACTGACTGCGTCTATTTAGAATATAAATGAGGT TGATGATGAGGACTTATCTGTCTCCTCATCACTGA |
| Sequence-based reagent | pbl_PL2 | This paper | SCRINSHOT padlock probe | /5Phos/ACCGGATCGTATTCAAAGATATATGAT TACTGACTGCGTCTATTTAGAATATAAATGAG GTTGATGATGAGGACTCTATCCGAGATGTCAATCAC TA |
| Sequence-based reagent | pbl_PL3 | This paper | SCRINSHOT padlock probe | /5Phos/CGAATAAACTATTTGAGACAGACAATG ATTACTGACTGCGTCTATTTAGAATATAAATGAGG TTGATGATGAGGACTCGGACTGGTGGTACAAT |

*Appendix 1 Continued on next page*

*Appendix 1 Continued*

| Reagent type (species) or resource | Designation | Source or reference | Identifiers | Additional information |
|---|---|---|---|---|
| Sequence-based reagent | AP-2alpha_PL1 | This paper | SCRINSHOT padlock probe | /5Phos/GCACTTGAGTTGTTGTTACTATCTATGAT TACTGACTGCGTCTATTTACTCCTACACTCTTCT TAAGTACCTTTCCTTCTTCGTGTTGTACACG |
| Sequence-based reagent | AP-2alpha_PL2 | This paper | SCRINSHOT padlock probe | /5Phos/TATGAATAACAGCTTACAGACATCTATGAT TACTGACTGCGTCTATTTACTCCTACACTCTTCT TAAGTACCTTTCCATGGCCGAGGAGAAA |
| Sequence-based reagent | AP-2alpha_PL3 | This paper | SCRINSHOT padlock probe | /5Phos/GCGACTGCAAATCATTCTTTACTAT GATTACTGACTGCGTCTATTTACTCCTACA CTCTTCTTAAGTACCTTTCGATTAACGTG CACAGGATTAC |
| Sequence-based reagent | CG11406_PL1 | This paper | SCRINSHOT padlock probe | /5Phos/AATTTGCCGCTATTGATGAGTGAT CCTCTATGATTACTGACTGCGTCTATTTAG ATAGTGGTCCACTGTCCTGGGTAGTCGTACTGCTGG |
| Sequence-based reagent | CG11406_PL2 | This paper | SCRINSHOT padlock probe | /5Phos/CCCGGCACCTGATACTGATCCTC TATGATTACTGACTGCGTCTATTTAGATAGTG GTCCACTGTCCTGGAACAGAAAGTCAATGTGA TTATAG |
| Sequence-based reagent | CG11406_PL3 | This paper | SCRINSHOT padlock probe | /5Phos/CTTATTCCTGTTTCCTCGTTCTGAT CCTCTATGATTACTGACTGCGTCTATTTAGA TAGTGGTCCACTGTCCTGGTCCACGTTTGGTGGAA |
| Sequence-based reagent | AkhR_PL1 | This paper | SCRINSHOT padlock probe | /5Phos/CAAGGATTTGCCTCAAAGATAATCCTCT ATGATTACTGACTGCGTCTATTTACTGAGGAGA ATGATCATCGTGATGTATCTGAATGCAACTGCAT |
| Sequence-based reagent | AkhR_PL2 | This paper | SCRINSHOT padlock probe | /5Phos/TCGCTTAATTGATCCAGTTGTCCTCTATG ATTACTGACTGCGTCTATTTACTGAGGAGAATGAT CATCGTGATTCCAGTTGAACATTAGCAGA |
| Sequence-based reagent | AkhR_PL3 | This paper | SCRINSHOT padlock probe | /5Phos/AGATGCATTAGCATGATATCAATACTCCT CTATGATTACTGACTGCGTCTATTTACTGAGGAG AATGATCATCGTGATGATCGGCGATGGCC |
| Sequence-based reagent | CG17186_PL1 | This paper | SCRINSHOT padlock probe | /5Phos/CAACATCTGGCTCTTCCTCTATGATTACT GACTGCGTCTATTTAGATAGAGGCTCTAGATTT CATCTACGTATCTATATCCTGTATATCCTCCA |
| Sequence-based reagent | CG17186_PL2 | This paper | SCRINSHOT padlock probe | /5Phos/GATCTATTTGAACCACCTCGTCTATGAT TACTGACTGCGTCTATTTAGATAGAGGCTCTAG ATTTCATCTACGTATATTCTATCATTTCAAAGCCGG |
| Sequence-based reagent | CG17186_PL3 | This paper | SCRINSHOT padlock probe | /5Phos/CAAGAGCTTCAGTTTCCTCTCTATGAT TACTGACTGCGTCTATTTAGATAGAGGCTCTAG ATTTCATCTACGTTCCAAGTATTCTACTGGCT |
| Sequence-based reagent | CG4872_PL1 | This paper | SCRINSHOT padlock probe | /5Phos/CTGCTCATATTGGCAATGAAATCTCTAT GATTACTGACTGCGTCTATTTAGAAACTATCCC AAGGAGCTAGAAATGTAGTAAGCTGGCGTAGAG |
| Sequence-based reagent | CG4872_PL2 | This paper | SCRINSHOT padlock probe | /5Phos/CGTGCGATTGTAAAGATTAAGCTCTATG ATTACTGACTGCGTCTATTTAGAAACTATCCCAA GGAGCTAGAAATCAGTGCCTTTGCCTG |
| Sequence-based reagent | CG4872_PL3 | This paper | SCRINSHOT padlock probe | /5Phos/CTCTCGGTGTTCAATTTCTGCTCTATGA TTACTGACTGCGTCTATTTAGAAACTATCCCAAG GAGCTAGAAATATACAGTAATCTGGGTTCCG |
| Sequence-based reagent | decay_PL1 | This paper | SCRINSHOT padlock probe | /5Phos/AAGAATTTGTCGAATGTGGAGATCCTCT ATGATTACTGACTGCGTCTATTTACTCAACCAGA TGAAGGAAATGCGTCAACGTTACGGAATGAG |
| Sequence-based reagent | decay_PL2 | This paper | SCRINSHOT padlock probe | /5Phos/GTTTGTTCTTGAGCGTCTTATCCTCTATG ATTACTGACTGCGTCTATTTACTCAACCAGATGAA GGAAATGCCTGGATGAAGAAGAGTTTGG |
| Sequence-based reagent | decay_PL3 | This paper | SCRINSHOT padlock probe | /5Phos/TCGTTGATCTCGGAGAAGATCCTCTATG ATTACTGACTGCGTCTATTTACTCAACCAGATGA AGGAAATGCCAACCTCTTTGAGCGTG |
| Sequence-based reagent | PH4alphaMP_PL1 | This paper | SCRINSHOT padlock probe | /5Phos/CCTCCATAAAGTCTAGGGATATCCTCTAT GATTACTGACTGCGTCTATTTACAATGTATGGATT CGTGAACGCCGATGATATTCCACCACTG |
| Sequence-based reagent | PH4alphaMP_PL2 | This paper | SCRINSHOT padlock probe | /5Phos/AAGTCGCGTAAGAGCTATCATCCTCTATG ATTACTGACTGCGTCTATTTACAATGTATGGATTC GTGAACGCCATCCAATTGATAAGTCTGCTG |

*Appendix 1 Continued on next page*

*Appendix 1 Continued*

| Reagent type (species) or resource | Designation | Source or reference | Identifiers | Additional information |
|---|---|---|---|---|
| Sequence-based reagent | PH4alphaMP_PL3 | This paper | SCRINSHOT padlock probe | /5Phos/CTAAGATTTCTTTGCTTGGATGATCCTCTAT GATTACTGACTGCGTCTATTTACAATGTATGGATTC GTGAACGCCTCTTCCTTAAGCGACAC |
| Sequence-based reagent | GstE11_PL1 | This paper | SCRINSHOT padlock probe | /5Phos/AGACAGTGCTCCAAGCCTCTATGATCCTC TATGATTACTGACTGCGTCTATTTACTGGTGGATG CCCGTCAAGTAATCACCTTCAGCC |
| Sequence-based reagent | GstE11_PL2 | This paper | SCRINSHOT padlock probe | /5Phos/CGACAATGAAACGGATTCGCTCTATGATC CTCTATGATTACTGACTGCGTCTATTTACTGGTGG ATGCCCGTCGAAATAGATCACCGGCT |
| Sequence-based reagent | GstE11_PL3 | This paper | SCRINSHOT padlock probe | /5Phos/ATATCCAGACCTTCCTGATTGCTCTATGAT CCTCTATGATTACTGACTGCGTCTATTTACTGGTG GATGCCCGTTTAACCAGTCCCACCAAC |
| Sequence-based reagent | Cog4_PL1 | This paper | SCRINSHOT padlock probe | /5Phos/GGATTCCAGTACTCCGATAGCCTCTATG ATTACTGACTGCGTCTATTTACAATCAGTTCCTG AACTACAGTCAGATGACATCTCTTTGTTATTCTCC |
| Sequence-based reagent | Cog4_PL2 | This paper | SCRINSHOT padlock probe | /5Phos/TGCACTATGAAGAACTGGACCTCTATGAT TACTGACTGCGTCTATTTACAATCAGTTCCTGAAC TACAGTCAGAGTTAAGTAGTCCATCCAAT |
| Sequence-based reagent | Cog4_PL3 | This paper | SCRINSHOT padlock probe | /5Phos/CTTCTCATAGCGTTCCATTATATCCCTCTATG ATTACTGACTGCGTCTATTTACAATCAGTTCCTGAAC TACAGTCACAGATCGGAGTTCTTCATGAT |
| Sequence-based reagent | Gr64c_PL1 | This paper | SCRINSHOT padlock probe | /5Phos/AAACTAAGAGATCCTATGGTATAGTGATCC TCTATGATTACTGACTGCGTCTATTTAGGCAAGGGA TACTTTCTGCTGATGCAGATCACGCTG |
| Sequence-based reagent | Gr64c_PL2 | This paper | SCRINSHOT padlock probe | /5Phos/GTTCTTGAATTGGAGATACGAGTGATCCT CTATGATTACTGACTGCGTCTATTTAGGCAAGGGA TACTTTCTGCTCGTGACACTTATGACTAAATGC |
| Sequence-based reagent | Gr64c_PL3 | This paper | SCRINSHOT padlock probe | /5Phos/GAAAGCTATGATAAATCTGCACTGATCCTC TATGATTACTGACTGCGTCTATTTAGGCAAGGGATA CTTTCTGCTCATAGTTGCCCTTGGATT |
| Sequence-based reagent | Idgf4_PL1 | This paper | SCRINSHOT padlock probe | /5Phos/CGTAGTCCAAGTTATTGATAATAGATGATCC TCTATGATTACTGACTGCGTCTATTTAGACTCAACG CAAACCTGTGGTAGGTGTGCAGGTTCA |
| Sequence-based reagent | Idgf4_PL2 | This paper | SCRINSHOT padlock probe | /5Phos/AACAAAGTAGATGATGAGATCCATGATCCTC TATGATTACTGACTGCGTCTATTTAGACTCAACGCA AACCTGTGAACTGTTGCCGTCATAGT |
| Sequence-based reagent | Idgf4_PL3 | This paper | SCRINSHOT padlock probe | /5Phos/GCTCTACAACTTGCTCTTAACATGATCCTCT ATGATTACTGACTGCGTCTATTTAGACTCAACGCAA ACCTGTGAAAGAATCAGAAGGATGAGGA |
| Sequence-based reagent | CG17477_PL1 | This paper | SCRINSHOT padlock probe | /5Phos/CCTGAGTTAGTGCGTTAAAGATCCTCTATGA TTACTGACTGCGTCTATTTAGATAATTACGGCTGGTC ACTGTATGTGGGTAGTTCAACCG |
| Sequence-based reagent | CG17477_PL2 | This paper | SCRINSHOT padlock probe | /5Phos/GGCAGACATCATGGACTATCCTCTATGATTA CTGACTGCGTCTATTTAGATAATTACGGCTGGTCACT GTCAGTTCCAGATCCTCGTA |
| Sequence-based reagent | CG17477_PL3 | This paper | SCRINSHOT padlock probe | /5Phos/TAAGCACAAATATGGCAAGGATCCTCTATGAT TACTGACTGCGTCTATTTAGATAATTACGGCTGGTCA CTGTTCCAATATTTGCCTGACGA |
| Sequence-based reagent | CG11069_PL1 | This paper | SCRINSHOT padlock probe | /5Phos/AAATCGGAAATAAAGATATGGTAGCCTCTATG ATTACTGACTGCGTCTATTTACAACTCTACTGATTGC CTACAACTGAGAACTTCTTGCGCC |
| Sequence-based reagent | CG11069_PL2 | This paper | SCRINSHOT padlock probe | /5Phos/CTCTTATGCGCCACTTGCCTCTATGATTACTG ACTGCGTCTATTTACAACTCTACTGATTGCCTACAAC TCACTGATATTAAGATACTCCACT |
| Sequence-based reagent | CG11069_PL3 | This paper | SCRINSHOT padlock probe | /5Phos/TGACCAACTGTATTCCAATAGCCTCTATGATT ACTGACTGCGTCTATTTACAACTCTACTGATTGCCTA CAACTCAATAGCATGACAGGATCTC |
| Sequence-based reagent | AdSL_PL1 | This paper | SCRINSHOT padlock probe | /5Phos/CGTCTTGAACGTTCTTCAGTGATCCTCTATGA TTACTGACTGCGTCTATTTACTGAGTCGACGTTCTT GATCTAGATCTAGATGTTCAGTTTGA |

*Appendix 1 Continued on next page*

Appendix 1 Continued

| Reagent type (species) or resource | Designation | Source or reference | Identifiers | Additional information |
|---|---|---|---|---|
| Sequence-based reagent | AdSL_PL2 | This paper | SCRINSHOT padlock probe | /5Phos/AGCCAGTAGCAACTGTTTTGATCCTCTATGATT ACTGACTGCGTCTATTTACTCGAGTCGACGTTCTTG AT CATAAGATATTCGGCTGCTTAA |
| Sequence-based reagent | AdSL_PL3 | This paper | SCRINSHOT padlock probe | /5Phos/CGAGACATTGCTGGTTAGTGATCCTCTATGATTA CTGACTGCGTCTATTTACTCGAGTCGACGTTCTTGA TG TGATAATCCATTTAGCAACAG |
| Sequence-based reagent | Tl_PL1 | This paper | SCRINSHOT padlock probe | /5Phos/GATCTGCACGTAGTCTTTGCTCTATGATTACTGA CTGCGTCTATTTACTACAACAATAATCTGGTGCATG TGT ATCCGTCAGATTGCACAT |
| Sequence-based reagent | Tl_PL2 | This paper | SCRINSHOT padlock probe | /5Phos/TTAGTTCGAGATGACTCAGATTCCTCTATGATTA CTGACTGCGTCTATTTACTACAACAATAATCTGGTG CAT GTGCATTTCCTCGATATTTGCCC |
| Sequence-based reagent | Tl_PL3 | This paper | SCRINSHOT padlock probe | /5Phos/TCCGAGTATATGATCACAATAATGCTCTATGATTA CTGACTGCGTCTATTTACTACAACAATAATCTGGTG CAT GTGCTTCTCCACATCTCCGATG |
| Sequence-based reagent | CG17283_PL1 | This paper | SCRINSHOT padlock probe | /5Phos/TTGGTCCTCACCTTTAATTTCATCCTCTATGATT ACTGACTGCGTCTATTTACAAGACCGAGTCAATCGT TA GAGACTGTCCTGCCTCTG |
| Sequence-based reagent | CG17283_PL2 | This paper | SCRINSHOT padlock probe | /5Phos/ATACGTAGTGGCAGCGATCCTCTATGATTACTG ACTGCGTCTATTTACAAGACCGAGTCAATCGTTAGA GT TAATACAGATAAGTCTAAGTGGA |
| Sequence-based reagent | CG17283_PL3 | This paper | SCRINSHOT padlock probe | /5Phos/GGTCACGAAAGTGGTTCATCCTCTATGATTACT GACTGCGTCTATTTACAAGACCGAGTCAATCGTTAG AC CTAGTATACCATCGAAATTTGA |
| Sequence-based reagent | Hr46_PL1 | This paper | SCRINSHOT padlock probe | /5Phos/GGTATATTGTTCAGCAGTGTATGATTACTGACTG CGTCTATTTAGAACATAATCGAATTTGCTAAGCTCA TAC GGAAATATCGCGGAAATTG |
| Sequence-based reagent | Hr46_PL2 | This paper | SCRINSHOT padlock probe | /5Phos/TCACTTTAACTGCTACTCAGATGATTACTGACT GCGTCTATTTAGAACATAATCGAATTTGCTAAGCTC AT ACGTATTCATTCGATTTCTCGC |
| Sequence-based reagent | Hr46_PL3 | This paper | SCRINSHOT padlock probe | /5Phos/GGCAAAGACAACCTAGGATGATTACTGACTG CGTCTATTTAGAACATAATCGAATTTGCTAAGCTCA TACCACATACTTTAGTTACAACGTAA |
| Sequence-based reagent | CG15117_PL1 | This paper | SCRINSHOT padlock probe | /5Phos/GTGGTTATGTCATTATAGGATGCTATGATTACTGACTGCGTCTATTTACAAACCCATTATCATGTCTGAATATGGTGATCCCTCAAATTGTCC |
| Sequence-based reagent | CG15117_PL2 | This paper | SCRINSHOT padlock probe | /5Phos/GATGTAGGTGAAGAGATCTTCCTATGATTACTGACTGCGTCTATTTACAAACCCATTATCATGTCTGAATATGGCATTAGGATATCAGATCGGC |
| Sequence-based reagent | CG15117_PL3 | This paper | SCRINSHOT padlock probe | /5Phos/AGCTATGGAGAAGTGCAGCTATGATTACTGACTGCGTCTATTTACAAACCCATTATCATGTCTGAATATGGCTTTGTTGACCAGGATCAG |
| Sequence-based reagent | gsb-n_PL1 | This paper | SCRINSHOT padlock probe | /5Phos/CCTCCAAGAATTCCATTTATGATGATTACTGACTGCGTCTATTTACAAATATAATTGTGCAATGGTATCGACTCCACTTATATCTGAGTCTCGC |

Appendix 1 Continued on next page

*Appendix 1 Continued*

| Reagent type (species) or resource | Designation | Source or reference | Identifiers | Additional information |
|---|---|---|---|---|
| Sequence-based reagent | gsb-n_PL2 | This paper | SCRINSHOT padlock probe | /5Phos/TCGAGAATCTAAGCGACGATGATTACTGAC TGCGTCTATTTACAAATATAATTGTGCAATGGTATC GACTCGAATGCAAAGAGCTTATAGCAA |
| Sequence-based reagent | gsb-n_PL3 | This paper | SCRINSHOT padlock probe | /5Phos/CGTTGCCAGCCTTAATGATGATTACTGACT GCGTCTATTTACAAATATAATTGTGCAATGGTATC GACTCATGTTCATAATGCATGTATGGAAA |
| Sequence-based reagent | shark_PL1 | This paper | SCRINSHOT padlock probe | /5Phos/TTCTCTGGTCAGAGTGCTCCTCTATGATT ACTGACTGCGTCTATTTATTTCGTATTTCGCAAG CTGATCTTTTAGTATGGCTACTGCCTC |
| Sequence-based reagent | shark_PL2 | This paper | SCRINSHOT padlock probe | /5Phos/AATTCTTCTACTGCCGTTTGTCCTCTATGA TTACTGACTGCGTCTATTTATTTCGTATTTCGCAA GCTGATCTAGCTTGTAGTTGAGCAGG |
| Sequence-based reagent | shark_PL3 | This paper | SCRINSHOT padlock probe | /5Phos/AACCGTTTGCACCAATTCTCCTCTATGATT ACTGACTGCGTCTATTTATTTCGTATTTCGCAAGC TGATCTAATGGAAACTGGCTTAAATGTG |
| Sequence-based reagent | Marf1_PL1 | This paper | SCRINSHOT padlock probe | /5Phos/GATTCGTCCAGCGAGTAGCTATGATTACT GACTGCGTCTATTTAATATATGTAATGGAACATTCC GATCCCAAGTAGAAAGCTGCGTATATGT |
| Sequence-based reagent | Marf1_PL2 | This paper | SCRINSHOT padlock probe | /5Phos/TGAATAGGTCTTTATGTGCTGCTATGATTAC TGACTGCGTCTATTTAATATATGTAATGGAACATTCC GATCCCGTTGGTCCGTTGTTAATAC |
| Sequence-based reagent | Marf1_PL3 | This paper | SCRINSHOT padlock probe | /5Phos/TTGGAGTCCTTCCATTAAGGCTATGATTACT GACTGCGTCTATTTAATATATGTAATGGAACATTCCG ATCCCGAATACTACAGTAAGGAACACTATG |
| Sequence-based reagent | cag_PL1 | This paper | SCRINSHOT padlock probe | /5Phos/CCCTCTTTAATCGACTGTTTGCCTCTATGAT TACTGACTGCGTCTATTTAAGTCTTCATCGCTCACT AGTAATGTTAAGTTCTCCACTCAACAGA |
| Sequence-based reagent | cag_PL2 | This paper | SCRINSHOT padlock probe | /5Phos/ACGGGAGAACCAATCAAAGCCTCTATGATTA CTGACTGCGTCTATTTAAGTCTTCATCGCTCACTAG TAATGGGTATATTCTCGGTACGGAC |
| Sequence-based reagent | cag_PL3 | This paper | SCRINSHOT padlock probe | /5Phos/GCTTGGGTCAGGGAAGCCTCTATGATTAC TGACTGCGTCTATTTAAGTCTTCATCGCTCACTAG TAATGGTATTTATATTGCACCTAAATTGCAT |
| Sequence-based reagent | CG7484_PL1 | This paper | SCRINSHOT padlock probe | /5Phos/TCTGTGTTCCACTTGGTTATCTCTATGAT TACTGACTGCGTCTATTTAGATCAAATACGTAAG AGGACTGGATGAAGAACTCCTCGACAGTG |
| Sequence-based reagent | CG7484_PL2 | This paper | SCRINSHOT padlock probe | /5Phos/ACTGAGGCTTGATGGTATCCTCTATGAT TACTGACTGCGTCTATTTAGATCAAATACGTAA GAGGACTGGATAAAGTGCAGCATTGCTTAC |
| Sequence-based reagent | CG7484_PL3 | This paper | SCRINSHOT padlock probe | /5Phos/GCGGATAGGCCCGCTCTATGATTACTGA CTGCGTCTATTTAGATCAAATACGTAAGAGGACT GGATCTTTGAATAAAGGCCTGAATCT |
| Sequence-based reagent | Rab3GAP1_PL1 | This paper | SCRINSHOT padlock probe | /5Phos/GAATGCAGTAGTGGAATTACTCCTCTATG ATTACTGACTGCGTCTATTTAGACCGACTGATAAG CAAGGATTACAGCTAGTTTGAGTATTGCT |
| Sequence-based reagent | Rab3GAP1_PL2 | This paper | SCRINSHOT padlock probe | /5Phos/CGTAGATCTGGACGAAGATTCCTCTAT GATTACTGACTGCGTCTATTTAGACCGACTGAT AAGCAAGGATTAGTATAGTAGTTCCATTTGGGAT |
| Sequence-based reagent | Rab3GAP1_PL3 | This paper | SCRINSHOT padlock probe | /5Phos/GTGCAACAGTTGTCATTGTCCTCTATGA TTACTGACTGCGTCTATTTAGACCGACTGATAA GCAAGGATTATCAGTCGTCGATTTACAATC |
| Sequence-based reagent | fz2_PL1 | This paper | SCRINSHOT padlock probe | /5Phos/AGGAAAGTGTTTGTGGAAGTATGATCCT CTATGATTACTGACTGCGTCTATTTAGCGTTTGT TCTACCTGCCGTTTCTCAGTCTATTTCGTTG |
| Sequence-based reagent | fz2_PL2 | This paper | SCRINSHOT padlock probe | /5Phos/TACAGGTAACATCCGATAACTATGTAT GATCCTCTATGATTACTGACTGCGTCTATTTA GCGTTTGTTCTACCTGCCTCAAAGTAGGCTGCTTCG |
| Sequence-based reagent | fz2_PL3 | This paper | SCRINSHOT padlock probe | /5Phos/TCAGTCGATTGTGTCTCATTTTATGAT CCTCTATGATTACTGACTGCGTCTATTTAGCG TTTGTTCTACCTGCCCCCAGGATCAGGACCT |

*Appendix 1 Continued on next page*

*Appendix 1 Continued*

| Reagent type (species) or resource | Designation | Source or reference | Identifiers | Additional information |
|---|---|---|---|---|
| Sequence-based reagent | fz_PL1 | This paper | SCRINSHOT padlock probe | /5Phos/ACTAAGAACTACGACTGCGCTATGAT TACTGACTGCGTCTATTTACTTGTTCTACGAG TACTACAACTTTGATAGATGATGAACTTTATACAAA GCC |
| Sequence-based reagent | fz_PL2 | This paper | SCRINSHOT padlock probe | /5Phos/AGATCGATATGGTGATGGGCTATGATT ACTGACTGCGTCTATTTACTTGTTCTACGAGTA CTACAACTTTGAGTCATGTTATATGGTATATTCTTGC |
| Sequence-based reagent | fz_PL3 | This paper | SCRINSHOT padlock probe | /5Phos/GCTAAACTCTAAGTAACTTTCGTTAC TATGATTACTGACTGCGTCTATTTACTTGTTC TACGAGTACTACAACTTTGAGCATAGGGAACGTCTA TGT |
| Sequence-based reagent | fz4_PL1 | This paper | SCRINSHOT padlock probe | /5Phos/TCATCGAGCCAAAGATCCCTATGATTA CTGACTGCGTCTATTTACAATCCTCTACTATAA CTTAATGCGGTCAAATAGCCGGAGATCAGAT |
| Sequence-based reagent | fz4_PL2 | This paper | SCRINSHOT padlock probe | /5Phos/CGAAGAATACGAATTCGATACGCTATG ATTACTGACTGCGTCTATTTACAATCCTCTACT ATAACTTAATGCGGTCTGATTTGTTGTTCTTTCGTTC |
| Sequence-based reagent | fz4_PL3 | This paper | SCRINSHOT padlock probe | /5Phos/GTCTAGAGTAGCGTCTCATTGCTATGA TTACTGACTGCGTCTATTTACAATCCTCTACTA TAACTTAATGCGGTGAAATGGTACCTAATCCAATTCC |
| Sequence-based reagent | fz3_PL1 | This paper | SCRINSHOT padlock probe | /5Phos/GGTTATGTAGGAGGCCAGGATCCTCTAT GATTACTGACTGCGTCTATTTACTCCAACTCTTT CGTATGCCACACAGGCTTAAATAGGAGGT |
| Sequence-based reagent | fz3_PL2 | This paper | SCRINSHOT padlock probe | /5Phos/GTGGTTTACGTATTTCTCCGGATCCTCTA TGATTACTGACTGCGTCTATTTACTCCAACTCTT TCGTATGCCATTTATATTACTGTGCGACTCTCA |
| Sequence-based reagent | fz3_PL3 | This paper | SCRINSHOT padlock probe | /5Phos/GTCTTCTTCTGAGAGCTCGGATCCTCTAT GATTACTGACTGCGTCTATTTACTCCAACTCTTTC GTATGCCACTAGAATGAGGGTCTCAGAC |
| Sequence-based reagent | spz_PL1 | This paper | SCRINSHOT padlock probe | /5Phos/TGTTACTGTTGCCTCTTATTTTCTATGATTA CTGACTGCGTCTATTTACGGTTATAAACGATATCA ATATCGCGCCTTTATACTGGTAGCTGG |
| Sequence-based reagent | spz_PL2 | This paper | SCRINSHOT padlock probe | /5Phos/CCACCGATCTTAAGTGTTTATAGTCTAT GATTACTGACTGCGTCTATTTACGGTTATAAAC GATATCAATATCGCGAAGCCCGATACCATCTG |
| Sequence-based reagent | spz_PL3 | This paper | SCRINSHOT padlock probe | /5Phos/TCATTCCTCAAAGGACGAGTCTATGATT ACTGACTGCGTCTATTTACGGTTATAAACGATAT CAATATCGCGAGGGATTGTGCTCTTTAGTG |
| Sequence-based reagent | spz4_PL1 | This paper | SCRINSHOT padlock probe | /5Phos/TCGTACTATTGGTATTTCCAGGTCTATGA TTACTGACTGCGTCTATTTACAGACGATTTGC GATTGTATTGAAATCGCAGTTCCTTTAGTAGTACTTA |
| Sequence-based reagent | spz4_PL2 | This paper | SCRINSHOT padlock probe | /5Phos/CAGGAGGGAATCTAATGGGTCTATGAT TACTGACTGCGTCTATTTACAGACGATTTGCGA TTGTATTGAAATCCTAATACTACTGTACAATTGTTCA |
| Sequence-based reagent | spz4_PL3 | This paper | SCRINSHOT padlock probe | /5Phos/ATCGTTATTAGTCTAGTGATCGTCTATGAT TACTGACTGCGTCTATTTACAGACGATTTGCGATT GTATTGAAATCAAATACCCGATAACCTCG |
| Sequence-based reagent | spz6_PL1 | This paper | SCRINSHOT padlock probe | /5Phos/TCCTCTAAAGGCTAGAACTTAGATGATCC TCTATGATTACTGACTGCGTCTATTTAGACGATA CCCTACCCAAGGCTTTCACCTCAAAGTTGTGT |
| Sequence-based reagent | spz6_PL2 | This paper | SCRINSHOT padlock probe | /5Phos/GCTCGTTTGCTTCTTGTAGATGATCCTCT ATGATTACTGACTGCGTCTATTTAGACGATACCCT ACCCAAGGCACTAGACGATAATCCATCTCC |
| Sequence-based reagent | spz6_PL3 | This paper | SCRINSHOT padlock probe | /5Phos/TTGTTTAGATCTCCGCAAATGATGATCCTC TATGATTACTGACTGCGTCTATTTAGACGATACCCT ACCCAAGGGATTCTTGGGTATCTGACCC |
| Sequence-based reagent | spz5_PL1 | This paper | SCRINSHOT padlock probe | /5Phos/AGGGTAAGACTAATGCTACTATGCCTCTATG ATTACTGACTGCGTCTATTTACCCACACATGAAACAC TCAATATCGATATCCATTAGATCGCTCAC |
| Sequence-based reagent | spz5_PL2 | This paper | SCRINSHOT padlock probe | /5Phos/CGGCTTGTAGATTTGTTTGCCTCTATGATTAC TGACTGCGTCTATTTACCCACACATGAAACACTCAAT ATCAATTCGGAAACCTAGTGTG |

*Appendix 1 Continued on next page*

*Appendix 1 Continued*

| Reagent type (species) or resource | Designation | Source or reference | Identifiers | Additional information |
|---|---|---|---|---|
| Sequence-based reagent | spz5_PL3 | This paper | SCRINSHOT padlock probe | /5Phos/AGCGAGACATCGCGCCTCTATGATTACTGA CTGCGTCTATTTACCCACACATGAAACACTCAATAT CGATATCCAAGAGATCCATGTTG |
| Sequence-based reagent | ham_det1_FAM | This paper | SCRINSHOT detection probe | GTAGCTCCUGATCCTGTAUATCAAGAT/36-FAM/ |
| Sequence-based reagent | ham_det2_FAM | This paper | SCRINSHOT detection probe | CCTTUAGAAAGCCAAAUGGTAGTTTC/36-FAM/ |
| Sequence-based reagent | ham_det3_FAM | This paper | SCRINSHOT detection probe | CTTCCTTAUGTCATCGAAGUACTCC/36-FAM/ |
| Sequence-based reagent | wnt2_det1_Cy3 | This paper | SCRINSHOT detection probe | GATTAACAUAAATGCAGCUCATTAGTGAC/3Cy3Sp/ |
| Sequence-based reagent | wnt2_det2_Cy3 | This paper | SCRINSHOT detection probe | CTTCAGCAUGAGTCTAUCGCC/3Cy3Sp/ |
| Sequence-based reagent | wnt2_det3_Cy3 | This paper | SCRINSHOT detection probe | AACGCATTUCCTATGACAUAGCC/3Cy3Sp/ |
| Sequence-based reagent | shg_det1_Cy5 | This paper | SCRINSHOT detection probe | CCGCTCAAUAATTATCAAUCGAATCGAA/3Cy5Sp/ |
| Sequence-based reagent | shg_det2_Cy5 | This paper | SCRINSHOT detection probe | CGCACGAUTATCGCAUACGAT/3Cy5Sp/ |
| Sequence-based reagent | shg_det3_Cy5 | This paper | SCRINSHOT detection probe | ACTCCTCCUTGGTTACGUTGAT/3Cy5Sp/ |
| Sequence-based reagent | shot_det1_Cy3 | This paper | SCRINSHOT detection probe | TAGTTGGUGCATAGUCGCG/3Cy3Sp/ |
| Sequence-based reagent | shot_det2_Cy3 | This paper | SCRINSHOT detection probe | GTTTCCTCUAGATCCTTUAGGGTCTT/3Cy3Sp/ |
| Sequence-based reagent | shot_det3_Cy3 | This paper | SCRINSHOT detection probe | GAATTTCGUATTGAGTAUGCCATCGA/3Cy3Sp/ |
| Sequence-based reagent | pbl_det | This paper | SCRINSHOT detection probe | GAATAUAAATGAGGUTGATGAUGAGGACT /FITC/ |
| Sequence-based reagent | AP-2alpha_det | This paper | SCRINSHOT detection probe | CTCCUACACTCUTCTTAAGUACCTTTC /CY3/ |
| Sequence-based reagent | CG11406_det | This paper | SCRINSHOT detection probe | GATAGUGGTCCACTGUCCTG /CY5/ |
| Sequence-based reagent | AkhR_det | This paper | SCRINSHOT detection probe | CUGAGGAGAAUGATCATCGUGAT /FITC/ |
| Sequence-based reagent | CG17186_det | This paper | SCRINSHOT detection probe | GAUAGAGGCTCUAGATTTCAUCTACG /CY3/ |
| Sequence-based reagent | CG4872_det | This paper | SCRINSHOT detection probe | GAAACTAUCCCAAGGAGCUAGAAAT /CY5/ |
| Sequence-based reagent | decay_det | This paper | SCRINSHOT detection probe | CUCAACCAGAUGAAGGAAAUGC /FITC/ |
| Sequence-based reagent | PH4alphaMP_det | This paper | SCRINSHOT detection probe | CAATGTAUGGATTCGUGAACGC /CY3/ |
| Sequence-based reagent | GstE11_det | This paper | SCRINSHOT detection probe | CTGGTGGAUGCCCGT /CY5/ |
| Sequence-based reagent | Cog4_det | This paper | SCRINSHOT detection probe | CAATCAGUTCCTGAACUACAGTCA /FITC/ |
| Sequence-based reagent | Gr64c_det | This paper | SCRINSHOT detection probe | GGCAAGGGAUACTTTCUGCT /CY3/ |
| Sequence-based reagent | Idgf4_det | This paper | SCRINSHOT detection probe | GACUCAACGCAAACCUGTG /CY5/ |
| Sequence-based reagent | CG17477_det | This paper | SCRINSHOT detection probe | GATAAUTACGGCUGGTCACUGT /CY3/ |
| Sequence-based reagent | CG11069_det | This paper | SCRINSHOT detection probe | CAACTCTACUGATTGCCUACAACT /FITC/ |

*Appendix 1 Continued on next page*

*Appendix 1 Continued*

| Reagent type (species) or resource | Designation | Source or reference | Identifiers | Additional information |
| --- | --- | --- | --- | --- |
| Sequence-based reagent | AdSL_det | This paper | SCRINSHOT detection probe | CTCGAGUCGACGTTCTUGAT /FITC/ |
| Sequence-based reagent | Tl_det | This paper | SCRINSHOT detection probe | CTACAACAAUAATCTGGUGCATGTG /CY3/ |
| Sequence-based reagent | CG17283_det | This paper | SCRINSHOT detection probe | CAAGACCGAGUCAATCGTUAGA /FITC/ |
| Sequence-based reagent | Hr46_det | This paper | SCRINSHOT detection probe | GAACATAAUCGAATTTGCUAAGCTCAUAC /CY3/ |
| Sequence-based reagent | CG15117_det | This paper | SCRINSHOT detection probe | CAAACCCAUTATCATGTCUGAATAUGG /CY5/ |
| Sequence-based reagent | gsb-n_det | This paper | SCRINSHOT detection probe | CAAATATAAUTGTGCAAUGGTATCGACUC /FITC/ |
| Sequence-based reagent | shark_det | This paper | SCRINSHOT detection probe | TTTCGTATUTCGCAAGCUGATCT /CY3/ |
| Sequence-based reagent | Marf1_det | This paper | SCRINSHOT detection probe | ATATATGUAATGGAACAUTCCGAUCCC /CY5/ |
| Sequence-based reagent | cag_det | This paper | SCRINSHOT detection probe | AGTCTTCAUCGCTCACUAGTAATG /CY3/ |
| Sequence-based reagent | CG7484_det | This paper | SCRINSHOT detection probe | GATCAAAUACGUAAGAGGACUGGAT /FITC/ |
| Sequence-based reagent | Rab3GAP1_det | This paper | SCRINSHOT detection probe | GACCGACUGAUAAGCAAGGAUTA /CY3/ |
| Sequence-based reagent | fz2_det | This paper | SCRINSHOT detection probe | GCGTTTGUTCTACCUGCC /FITC/ |
| Sequence-based reagent | fz_det | This paper | SCRINSHOT detection probe | CTTGTTCUACGAGTACUACAACTUTGA /FITC/ |
| Sequence-based reagent | fz4_det | This paper | SCRINSHOT detection probe | CAATCCTCUACTATAACUTAATGCGGT /CY3/ |
| Sequence-based reagent | fz3_det | This paper | SCRINSHOT detection probe | CTCCAACUCTTTCGTAUGCCA /CY5/ |
| Sequence-based reagent | spz _det | This paper | SCRINSHOT detection probe | CGGTTAUAAACGATAUCAATAUCGCG /CY5/ |
| Sequence-based reagent | spz4_det | This paper | SCRINSHOT detection probe | CAGACGATUTGCGATUGTATUGAAAT /CY3/ |
| Sequence-based reagent | spz6_det | This paper | SCRINSHOT detection probe | GACGAUACCCUACCCAAGG /CY5/ |
| Sequence-based reagent | spz5_det | This paper | SCRINSHOT detection probe | CCCACACAUGAAACACUCAATATC /FITC/ |
| Commercial assay or kit | SuperSignal West Femto Maximum Sensitivity Substrate | Thermo Fisher | Cat# 34096 | |
| Commercial assay or kit | Effectene Transfection kit | QIAGEN | Cat# 301427 | |
| Commercial assay or kit | Dual-Glo Luciferase Assay System | Promega | Cat# E2940 | |

