## [Editor Report · eLife Assessment]

This **important** study addresses an essential morphogenetic process-epithelial fusion-by identifying the transcription factor Hamlet as a potential master regulator. Using a combination of genetic, cell biological, and omics approaches, including a comprehensive RNAi screen and high-quality imaging, the authors provide **compelling** evidence for Hamlet's role in coordinating cell fate and differentiation. The findings are robust and of broad interest to developmental biologists and geneticists.

---

## [Referee Report · Reviewer #1 (Public review)]

Summary:

Wang et al. identify Hamlet, a PR-containing transcription factor, as a master regulator of reproductive development in *Drosophila*. Specifically, the fusion between the gonad and genital disc that is necessary for development of a continuous testes and seminal vesicle tissue essential for fertility. To do so, the authors generate novel Hamlet null mutants by CRISPR/Cas9 gene editing and characterize the morphological, physiological, and gene expression changes of the mutants using immunofluorescence, RNA-seq, cut-tag, and in-situ analysis. Thus, Hamlet is discovered to regulate a unique expression program, which includes Wnt2 and Tl, that is necessary for testis development and fertility.

Strengths:

This is a rigorous and comprehensive study that identifies the Hamlet dependent gene expression program mediating reproductive development in *Drosophila*. The Hamlet transcription targets are further characterized by Gal4/UAS-RNAi confirming their role in reproductive development. Finally, the study points to a role for Wnt2 and Tl as well as other Hamlet transcriptionally regulated genes in epithelial tissue fusion.

Weaknesses:

None noted.

---

## [Referee Report · Reviewer #2 (Public review)]

Strengths:

Wang and colleagues successfully uncovered an important function of the *Drosophila* PRDM16/PRDM3 homolog Hamlet (Ham) - a PR domain containing transcription factor with known roles in the nervous system in *Drosophila*. To do so, they generated and analyzed new mutants lacking the PR domain, and also employed diverse preexisting tools. In doing so, they made a fascinating discovery: They found that PR-domain containing isoforms of ham are crucial in the intriguing development of the fly genital tract. Wang and colleagues found three distinct roles of Ham: (1) Specifying the position of the testis terminal epithelium within the testis, (2) allowing normal shaping and growth of the anlagen of the seminal vesicles and paragonia and (3) enabling the crucial epithelial fusion between the seminal vesicle and the testis terminal epithelium. The mutant blocks fusion even if the parts are positioned correctly. The last finding is especially important, as there are few models allowing one to dissect the molecular underpinnings of heterotypic epithelial fusion in development. Their data suggest that they found a master regulator of this collective cell behavior. Further, they identified some of the cell biological players downstream of Ham, like for example E-Cadherin and Crumbs. In a holistic approach, they performed RNAseq and intersected them with the CUT&TAG-method, to find a comprehensive list of downstream factors directly regulated by Ham. Their function in the fusion process was validated by a tissue-specific RNAi screen. Meticulously, Wang and colleagues performed multiplexed in situ hybridization and analyzed different mutants, to gain a first understanding of the most important downstream-pathways they characterized - which are Wnt2 and Toll.

This study pioneers a completely new system. It is a model for exploring a process crucial in morphogenesis across animal species, yet not well-understood. Wang and colleagues not only identified a crucial regulator of heterotypic epithelial fusion but took on the considerable effort of meticulously pinning down functionally important downstream effectors by using many state-of-the-art methods. This is especially impressive, as dissection of pupal genital discs before epithelial fusion is a time-consuming and difficult task. This promising work will be the foundation future studies build on, to further elucidate how this epithelial fusion works, for example on a cell biological and biomechanical level.

Weaknesses:

The developing testis-genital disc system has many moving parts. Myotube migration was previously shown to be crucial for testis shape. This means, that there is the potential of non-tissue autonomous defects upon knockdown of genes in the genital disc or the terminal epithelium, affecting myotube behavior which in turn affects epithelial fusion, as myotubes might create the first "bridge" bringing the two epithelia together. Nevertheless, this is outside the scope of this work and could be addressed in the future.

---

## [Author Response]

The following is the authors’ response to the original reviews

**Reviewer #1 (Public review):**
Summary:Wang et al. identify Hamlet, a PR-containing transcription factor, as a master regulator of reproductive development in *Drosophila*. Specifically, the fusion between the gonad and genital disc is necessary for the development of continuous testes and seminal vesicle tissue essential for fertility. To do this, the authors generate novel Hamlet null mutants by CRISPR/Cas9 gene editing and characterize the morphological, physiological, and gene expression changes of the mutants using immunofluorescence, RNA-seq, cut-tag, and in-situ analysis. Thus, Hamlet is discovered to regulate a unique expression program, which includes Wnt2 and Tl, that is necessary for testis development and fertility.Strengths:This is a rigorous and comprehensive study that identifies the Hamlet-dependent gene expression program mediating reproductive development in *Drosophila*. The Hamlet transcription targets are further characterized by Gal4/UAS-RNAi confirming their role in reproductive development. Finally, the study points to a role for Wnt2 and Tl as well as other Hamlet transcriptionally regulated genes in epithelial tissue fusion.

We appreciate that the reviewer thinks our study is rigorous.

Weaknesses:The image resolution and presentation of figures is a major issue in this study. As a nonexpert, it is nearly impossible to see the morphological changes as described in the results. Quantification of all cell biological phenotypes is also lacking therefore reducing the impact of this study to those familiar with tissue fusion events in *Drosophila* development.

In the revised version, we have improved the image presentation and resolution. For all the images with more than two channels, we included single-channel images, changed the green color to lime and the red to magenta, highlighted the testis (TE) and seminal vescicles to make morphological changes more visible.

We had quantification for marker gene expression in the original version, and now also included quantification for cell biological phenotypes which are generally with 100% penetrance.

**Reviewer #2 (Public review):**
Strengths:Wang and colleagues successfully uncovered an important function of the *Drosophila* PRDM16/PRDM3 homolog Hamlet (Ham) - a PR domain-containing transcription factor with known roles in the nervous system in *Drosophila*. To do so, they generated and analyzed new mutants lacking the PR domain, and also employed diverse preexisting tools. In doing so, they made a fascinating discovery: They found that PR-domain containing isoforms of ham are crucial in the intriguing development of the fly genital tract. Wang and colleagues found three distinct roles of Ham: (1) specifying the position of the testis terminal epithelium within the testis, (2) allowing normal shaping and growth of the anlagen of the seminal vesicles and paragonia and (3) enabling the crucial epithelial fusion between the seminal vesicle and the testis terminal epithelium. The mutant blocks fusion even if the parts are positioned correctly. The last finding is especially important, as there are few models allowing one to dissect the molecular underpinnings of heterotypic epithelial fusion in development. Their data suggest that they found a master regulator of this collective cell behavior. Further, they identified some of the cell biological players downstream of Ham, like for example E-Cadherin and Crumbs. In a holistic approach, they performed RNAseq and intersected them with the CUT&TAG-method, to find a comprehensive list of downstream factors directly regulated by Ham. Their function in the fusion process was validated by a tissue-specific RNAi screen. Meticulously, Wang and colleagues performed multiplexed in situ hybridization and analyzed different mutants, to gain a first understanding of the most important downstream pathways they characterized, which are Wnt2 and Toll.This study pioneers a completely new system. It is a model for exploring a process crucial in morphogenesis across animal species, yet not well understood. Wang and colleagues not only identified a crucial regulator of heterotypic epithelial fusion but took on the considerable effort of meticulously pinning down functionally important downstream effectors by using many state-of-the-art methods. This is especially impressive, as the dissection of pupal genital discs before epithelial fusion is a time-consuming and difficult task. This promising work will be the foundation future studies build on, to further elucidate how this epithelial fusion works, for example on a cell biological and biomechanical level.

We appreciate that the reviewer thinks our study is orginal and important.

Weaknesses:The developing testis-genital disc system has many moving parts. Myotube migration was previously shown to be crucial for testis shape. This means, that there is the potential of non-tissue autonomous defects upon knockdown of genes in the genital disc or the terminal epithelium, affecting myotube behavior which in turn affects fusion, as myotubes might create the first "bridge" bringing the epithelia together. The authors clearly showed that their driver tools do not cause expression in myoblasts/myotubes, but this does not exclude non-tissue autonomous defects in their RNAi screen. Nevertheless, this is outside the scope of this work.

We thank the reviewer’s consideration of non-tissue autonomous defects upon gene knockdown. The driver, hamRSGal4, drives reporter gene expression mainly in the RS epithelia, but we did observe weak expression of the reporter in the myoblasts before they differentiate into myotubes. Thus, we could not rule out a non-tissue autonomou effect in the RNAi screen. So we now included a statement in the result, “Given that the hamRSGal4 driver is highly expressed in the TE and SV epithelia, we expect highly effective knockdown occurs only in these epithelial cells. However, hamRSGal4 also drives weak expression in the myoblasts before they differentiated into myotubes (Supplementary Fig. 5B), which may result in a non-tissue autonomous effect when knocking down the candidate genes expressed in myoblasts.”

However, one point that could be addressed in this study: the RNAseq and CUT&TAG experiments would profit from adding principal component analyses, elucidating similarities and differences of the diverse biological and technical replicates.

Thanks for the suggestion. We now have included the PCA analyses in supplementary figure 6A-B and the corresponding description in the text. The PCA graphs validated the consistency between biological replicates of the RNA-seq samples. The Cut&Tag graphs confirm the consistency between the two biological replicates from the GFP samples, but show a higher variability between the w1118 replicates. Importantly, we only considered the overlapped peaks pulled by the GFP antibody from the ham_GFP genotype and the Ham antibody from the wildtype (w1118) sample as true Ham binding sites.

**Recommendations for the authors:**

**Reviewer #1 (Recommendations for the authors):**
Major Concern:(1) The image resolution and presentation of figures (Figures 2, 5, 6, and 7) is a major issue in this study. As a non-expert, it is nearly impossible to see the morphological changes as described in the results. Images need to be captured at higher resolution and zoomed in with arrows denoting changes as described. Individual channels, particularly for intensity measurement need to be shown in black and white in addition to merged images. Images also need pseudo-colored for color-blind individuals (i.e. no red-green staining).

The images were captured at a high resolution, but somehow the resolution was drammaticlly reduced in the BioRxiv PDF. We try to overcome this by directly submitting the PDF in the Elife submission system. In the revised version, we have included single-channel images, changed the green and red colors to lime and magenta for color blindness. We also highlighted the testis (TE) and seminal vescicle structures in the images to make morphological changes more visible.

(2) The penetrance of morphological changes observed in RT development is also unclear and needs to be rigorously quantified for data in Figures 2, 5, and 7.

We now included quantification for cell biological phenotypes which are generally with 100% penetrance. The percentage of the penetrance and the number of animals used are indicated in each corresponding image.

**Reviewer #2 (Recommendations for the authors):**
Major Points(1) Lines 193- 220 I would strongly suggest pointing out the obvious shape defects of the testes visible in Figure 2A ("Spheres" instead of "Spirals"). These are probably a direct consequence of a lack in the epithelial connection that myotubes require to migrate onto the testis (in a normal way) as depicted in the cartoons, allowing the testis to adopt a spiral shape through myotube-sculpting (Bischoff et al., 2021), further confirming the authors' findings!

Good point. In the revised text, we have added more description of the testis shape defects and pointed out a potential contribution from compromised myotube migration.

(2) Line 216: "Often separated from each other". Here it would be important to mention how often. If the authors cannot quantify that from existing data, I suggest carrying it out in adult/pharate adult genital tracts (if there is no strong survivor bias due to the lethality of stronger affected animals), as this is much easier than timing prepupae. This should be a quick and easy experiment.

Because it is hard to tell whether the separation of the SV and TE was caused by developmental defects or sometimes could be due to technical issues (bad dissection), we now change the description to, “control animals always showed connected TE and SV, whereas ham mutant TE and SV tissues were either separated from each other, or appeared contacted but with the epithelial tubes being discontinuous (Fig. 2B).” Additionally, we quantified the disconnection phenotype, which is 100% penetrance in 18 mutant animals. This quantification is now included in the figure.

(3) Lines 289-305, Figure 3. I could only find how many replicates were analyzed in the RNAseq/CUT&Tag experiments in the Material & Methods section. I would add that at least in the figure legends, and perhaps even in the main text. Most importantly, I would add a Principal Component Analysis (one for RNAseq and one for the CUT&TAG experiment), to demonstrate the similarity of biological replicates (3x RNaseq, 4x Cut&Tag) but also of the technical replicates (RNAseq: wt & wt/dg, ham/ham & ham/df, GD & TE; CUT&TAG: Antibody & GFP-Antibody, TG&TE...). This should be very easy with the existing data, and clearly demonstrate similarities & differences in the different types of replicates and conditions.

Principle component analysis and its description are now added to Supplementary Fig 6 and the main text respectively.

(4) Line 321; Supplementary Table 1: In the table, I cannot find which genes are down- or upregulated - something that I think is very important. I would add that, and remove the "color" column, which does not add any useful information.

In Supplementary table 1, the first sheet includes upregulated genes while the second sheet includes downregulated genes. We removed the column “color” as suggested.

(5) Line 409: SCRINSHOT was carried out with candidate genes from the screen. One gene I could not find in that list was the potential microtubule-actin crosslinker shot. If shot knockdown caused a phenotype, then I would clearly mention and show it. If not, I would mention why a shot is important, nonetheless.

shot is one of the candidate target genes selected from our RNA-seq and Cut&Tag data. However, in the RNAi screen, knocking down shot with the available RNAi lines did not cause any obvious phenotype. These could be due to inefficient RNAi knockdown or redundancy with other factors. We anyway wanted to examine shot expression pattern in the developing RS, give the important role of shot in epithelial fusion (Lee S., 2002). Using SCRINSHOT, we could detect epithelial-specific expression of shot, implying its potential function in this context. We now revised the text to clarify this point.

Minor points(1) Cartoons in Figure 1: The cartoons look like they were inspired by the cartoon from Kozopas et al., 1998 Fig. 10 or Rothenbusch-Fender et al., 2016 Fig 1. I think the manuscript would greatly profit from better cartoons, that are closer to what the tissue really looks like (see Figure 1H, 2G), to allow people to understand the somewhat complicated architecture. The anlagen of the seminal vesicles/paragonia looks like a butterfly with a high columnar epithelium with a visible separation between paragonia/seminal vesicles (upper/lower "wing" of the "butterfly"). Descriptions like "unseparated" paragonia/seminal vesicle anlagen, would be much easier to understand if the cartoons would for example reflect this separation. It would even be better to add cartoons of the phenotypic classes too, and to put them right next to the micrographs. (Another nitpick with the cartoons: pigment cells are drastically larger and fewer in number (See: Bischoff et al., 2021 Figure 1E & MovieM1).)

Thanks for the suggestion. We have updated Figure 1 by adding additional illustrations showing the accessory gland and seminal vesicle structures in the pupal stage and changing the size of pigment cells.

(2) Line 95-121 I would also briefly introduce PR domains, here.

We have added a brief descripition of the PR domains.

(3) Line 152, 158, 160, 162. When first reading it, I was a bit confused by the usage of the word sensory organ. I would at least introduce that bristles are also known as external mechanosensory organs.

We have now revised the description to “mechano-sensory organ”.

eg. Line 184, 194, and many more. Most times, the authors call testis muscle precursors "myoblasts". This is correct sometimes, but only when referring to the stage before myoblast-fusion, which takes place directly before epithelial fusion (28 h APF). Postmyoblast-fusion (eg. during migration onto the testis), these cells should be called myotubes or nascent myotubes, as the fly muscle community defined the term myoblast as the singlenuclei precursors to myotubes.

We have now revised the description accordingly.

(4) Line 217/Figure 2B. It looks like there is a myotube bridge between the testis and the genital disc. I would point that out if it's true. If the authors have a larger z-stack of this connection, I suggest creating an MIP, and checking if there are little clusters of two/three/four nuclei packed together. This would clearly show that the cells in between are indeed myotubes (granted that loss of ham does not introduce myoblast-fusion-defects).

We do not have a Z-stack of this connection, and thus can not confirm whether the cells in this image are myotubes. However, we found that mytubes can migrate onto the testis and form the muscular sheet in the ham mutant despite reduced myotube density. At the junction there are myotubes, suggesting that loss of ham does not introduce myoblast-fusion defects. These results are now included in the revised manuscript, supplementary Fig. 5 C-D.

(5) Line 231/Supplementary Fig. 3C-G: I would add to the cartoons, where the different markers are expressed.

We have added marker gene expression in the cartoons.

(6) Line 239. I don't see what Figure 1A/1H refers to, here. I would perhaps just remove it.

Yes, we have removed it.

(7) Line 232. I would rephrase the beginning of the sentence to: Our data suggest Ham to be...

Yes, we have revised it.

(8) Line 248-250/Figure 2F. Clonal analyses are great, but I think single channels should be shown in black and white. Also, a version without the white dashed line should be shown, to clearly see the differences between wt and ham-mutant cells.

Now single channel images from the green and red images are presented in Supplementary Figures. This particular one is in Supplementary Figure 3B.

(9) Line 490. The Toll-9 phenotype was identified on the sterility effect/lack-of-spermphenotype alone, and it was deduced, that this suggests connection defects. By showing the right focus plane in Fig S8B (lower right), it should be easy to directly show whether there is a connection defect or not. Also, one would expect clearer testis-shaping defects, like in ham-mutants, as a loss of connection should also affect myotube migration to shape the testis. This is just a minor point, as it only affects supplementary data with no larger impact on the overall findings, even if Toll-9 is shown not to have a defect, after all.

We find that scoring defects at the junction site at the adult stage is difficult and may not be always accurate. Instead, we score the presence of sperms in the SV, which indirectly but firmly suggests successful connection between the TE and SV. We have now included a quantification graph, showing the penetrance of the phentoype in the new Supplementary Fig.14C. There were indeed morphological defects of TE in Toll-9 RNAi animals. We now included the image and quantification in the new Supplementary Fig.14B.